# Scalable multifunctional MOFs-textiles via diazonium chemistry

Wulong Li [1,2], Zhen Yu[3], Yaoxin Zhang[4], Cun Lv[5], Xiaoxiang He[5], Shuai Wang[1], Zhixun Wang [1], Bing He[1], Shixing Yuan[1], Jiwu Xin[1], Yanting Liu[1], Tianzhu Zhou[1], Zhanxiong Li [5,6] ✉, Swee Ching Tan [2] ✉ & Lei Wei [1] ✉

Cellulose fiber-based textiles are ubiquitous in daily life for their processability, biodegradability, and outstanding flexibility. Integrating cellulose textiles with functional coating materials can unlock their potential functionalities to engage diverse applications. Metal-organic frameworks (MOFs) are ideal candidate materials for such integration, thanks to their unique merits, such as large specific surface area, tunable pore size, and species diversity. However, achieving scalable fabrication of MOFs-textiles with high mechanical durability remains challenging. Here, we report a facile and scalable strategy for direct MOF growth on cotton fibers grafted via the diazonium chemistry. The as-prepared ZIF-67-Cotton textile (ZIF-67-CT) exhibits excellent ultraviolet (UV) resistance and organic contamination degradation via the peroxymonosulfate activation. The ZIF-67-CT is also used to encapsulate essential oils such as carvacrol to enable antibacterial activity against *E. coli* and *S. aureus*. Additionally, by directly tethering a hydrophobic molecular layer onto the MOF-coated surface, superhydrophobic ZIF-67-CT is achieved with excellent self-cleaning, antifouling, and oil-water separation performances. More importantly, the reported strategy is generic and applicable to other MOFs and cellulose fiber-based materials, and various large-scale multi-functional MOFs-textiles can be successfully manufactured, resulting in vast applications in wastewater purification, fragrance industry, and outdoor gears.

Metal-organic frameworks, featured with tunable pore sizes, large specific surface areas, and rich chemical functionalities, can easily change their structures and properties with different metal ions and organic ligands under appropriate reaction conditions[1–3]. They exhibit tremendous application potential in gas separation, catalyst, air water harvesting, energy storage, drug delivery, etc[2,4–6]. However, the powder form of MOFs limits their extending applications due to poor collectability and processability. While, textiles are widely usable due to their excellent processability, flexibility, and mechanical strength, especially for cellulose fiber-based textiles. Therefore, the integration of MOFs with textiles endows the formation of MOFs-textiles with the merits of textiles and MOFs[7–11]. In particular, MOFs could advance beyond their pristine properties in powder form and be tailored into different shapes to fit the expected applications, such as water sorption devices, catalyzing devices, and personal protective equipment. As a result, MOFs-textiles could extend and promote their potential applications in healthcare, safety, and environmental protection[12–16].

[1]School of Electrical and Electronic Engineering, Nanyang Technological University, Singapore, Singapore. [2]Department of Materials Science and Engineering, National University of Singapore, Singapore, Singapore. [3]School of Environmental Science and Engineering, Tianjin Key Lab of Biomass/Wastes Utilization, Tianjin University, Tianjin, China. [4]China-UK Low Carbon College, Shanghai Jiao Tong University, Shanghai, China. [5]College of Textile and Clothing Engineering, Soochow University, Suzhou, China. [6]National Engineering Laboratory for Modern Silk, Soochow University, Suzhou, China. ✉e-mail: lizhanxiong@suda.edu.cn; msetansc@nus.edu.sg; wei.lei@ntu.edu.sg

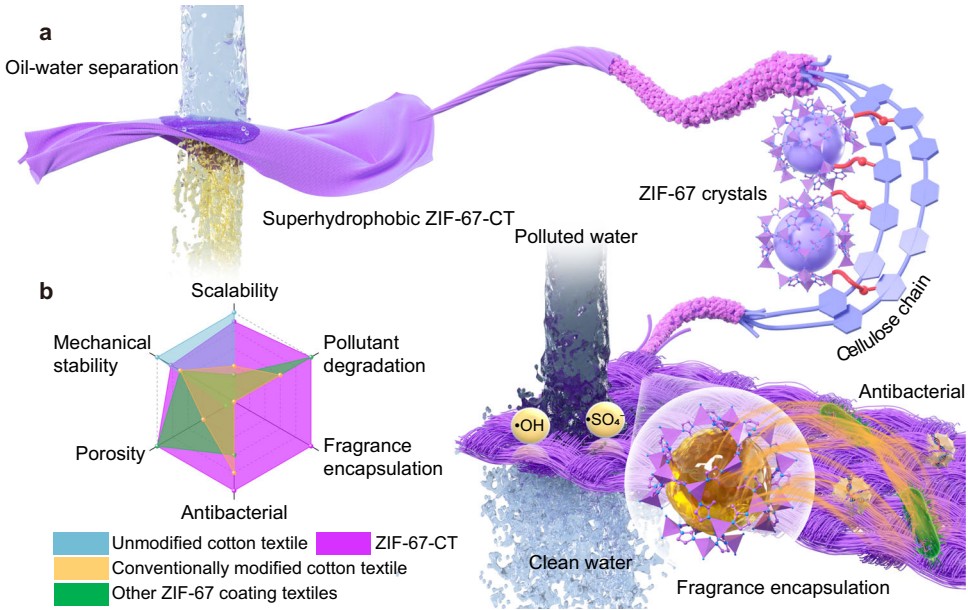

**Fig. 1 | The structure and functionality of ZIF-67-CT.** a Schematic displaying the structure of ZIF-67-CT and its organic pollutant degradation, fragrance encapsulation, and antibacterial properties, and the oil-water separation property of superhydrophobic ZIF-67-CT. The carboxymethylated cellulose fibers' surface can provide abundant coordinated sites for the cobalt ions and organic ligands in situ growth, forming dense MOF coating on the surface of fibers. **b** Comparison of the performance factors of ZIF-67-CT to those of unmodified, conventionally modified cotton textiles, and other ZIF-67 coating textiles (Supplementary Note 1).

Cellulose-based textiles are natural textiles, such as cotton and linen, composed of cellulose fibers that consist of cellulose molecules with hydroxyl groups. The cellulose chain is derived from D-glucose monomers, which could be bio-synthesized to form elementary fibril and then cellulose nanofibers by self-assembling, and ultimately microfibers[17]. To endow textiles with multiple functionalities, such as antibacterial, superhydrophobic, or flame-retardant properties[18–20], surface modification is an effective method. Particularly, diazonium chemistry has been extensively studied for surface modification, because the aryldiazonium salts with reactive groups could serve as surface modifiers or coupling agents for binding carbon nanotubes, macromolecules, nanoparticles, and functional materials[21,22] to substrates (for example, metal, semiconductors, and carbon-based nanostructures)[23–25]. This is achieved by a strong aryl-surface covalent bond, of which the energy is over 50 kcal/mol, high enough to ensure the structure's stability[22,26].

Recently, a range of MOFs (for example, MOF-808, ZIF-8, HKUST-1, MOF-5, and UiO-66-NH$_2$) have been physically incorporated into synthetic polymer fibers (for example, polypropylene, polyacrylonitrile, polystyrene, and nylon) and natural fibers (for example, cotton, wool, and silk fiber)[27–31]. However, current mainstream strategies for fabricating MOFs-fibers are typically incorporated into the fibers with non-covalent bonds using methods such as deposition dip-coating, hot-pressing, and spraying[32–34], which usually suffer from poor washing and wearing durability due to their low adhesivity, non-covalent bond force, and weak bonding ability (for example, van der Waals forces and electrostatic interactions). In addition, the complex fabrication process and difficulty in scaling up manufacturing limit them only in the lab preparation, not suitable for manufacturing on an industry scale[35–37]. Finally, most techniques only integrate one type of MOF with fibers, further limiting their widespread applications[14,38–40]. These critical challenges associated with scalable fabrication, mechanical durability, and substrate universality remain to be addressed.

Here, we report a generic strategy for preparing highly stable multifunctional MOFs-textiles via diazonium chemistry and in situ growth by soaking a cotton textile in 3-aminobenzoic acid diazonium salt HCl solution and metal-contenting and ligand-contenting solutions, respectively. Specifically, the covalent grafted carboxyl polymer chain brushes on the fiber surface create abundant carboxyl group sites, which assist the initial coordination of cobalt ions to the fiber surface and facilitate the subsequent in situ growth of MOF nanoparticles, forming a uniform and dense ZIF-67 MOF coating. Such a method allows us to scalable manufacture washable MOFs-textiles with UV resistance, antibacterial, fragrance encapsulation, and wastewater purification properties (Fig. 1), showing great potential in household products and industrial use. Besides, the as-prepared superhydrophobic MOFs-textiles can be widely used in outdoor products and public facilities for self-cleaning, antifouling, anti-icing, and oil-water separation properties. The coordination bonding between ZIF-67 and carboxyl chains of cellulose fibers endows this ZIF-67-textile to be highly stable in air and water. In addition, the fabrication of ZIF-67-CT is facile and scalable, and can also be applied to other cellulose-based fibers (for example, linen and paper) and MOFs (for example, ZIF-8, UiO-66-NH$_2$, and MOF-303). This strategy is expected to provide a new avenue for the fabrication of scalable multifunctional MOFs-textiles. More useful functions could hence be enabled for the practical applications of textiles, including improved environmental remediation, public health, and personal protective equipment.

## Results

### Synthesis and characterization of the ZIF-67-CT

The fabrication process of ZIF-67-CT requires two simple steps, as shown in Fig. 2a, and the detailed reaction mechanism is shown in Supplementary Figs. 1 and 2 and Supplementary Note 2. The first step is the carboxymethylation of cotton textiles via diazonium chemistry. One piece of CT was soaked in 3-aminobenzoic acid diazonium salt HCl solution with vitamin C and incubated for 12 h at room temperature. Then, the yellow textile was thoroughly washed and dried at room temperature, obtaining the carboxymethylated CT termed as CT-COOH. The second step is the ZIF-67 in situ growth on the fiber surface of CT-COOH. The yellow textile was soaked in the Co$^{2+}$-contenting methanol solution for 1 h, then the 2-methylimidazole (2-MeIm) methanol solution was added and incubated for 12 h at 30 °C. Then, the

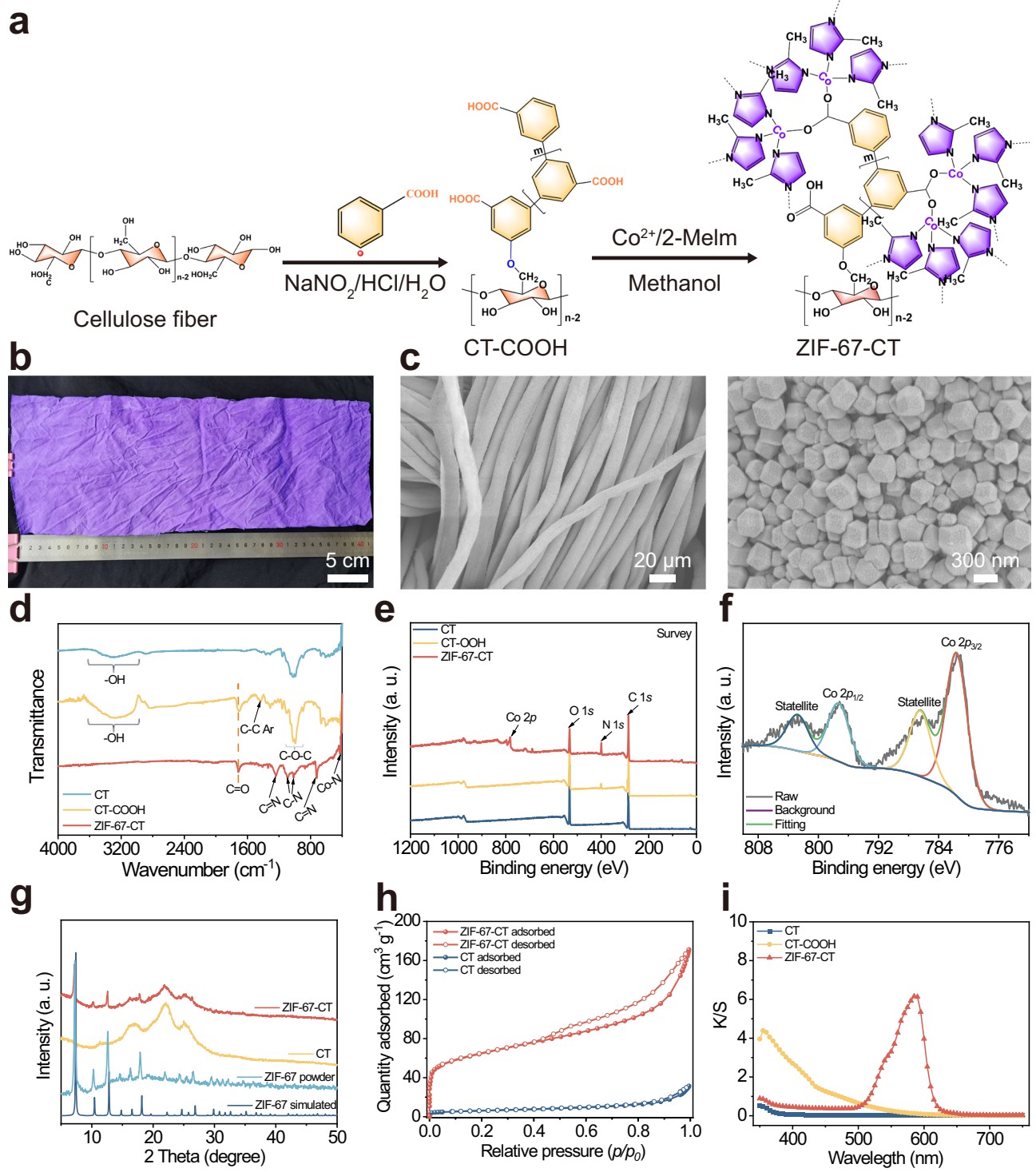

**Fig. 2 | Synthesis and characterization of the ZIF-67-CT. a** Synthetic route of ZIF-67-CT via the in situ growth. **b** Photograph of the ZIF-67-CT (42 cm × 15 cm). **c** SEM images of ZIF-67-CT with different magnifications. **d, e** The (**d**) ATR-FTIR and (**e**) XPS spectra of CT, CT-COOH, and ZIF-67-CT. **f** The high-resolution XPS Co 2*p* spectrum of the ZIF-67-CT. **g** The XRD patterns of CT, ZIF-67-CT, ZIF-67 powder, and simulated ZIF-67. **h** N₂ adsorption–desorption isotherms of CT and ZIF-67-CT. **i** The color absorption spectra of CT, CT-COOH, and ZIF-67-CT.

purple textile was thoroughly washed with methanol to remove residual nonbonded MOF particles and chemicals, and dried for further use (see Methods for more details). Photographs of CT, CT-COOH, and ZIF-67-CT (42 cm × 15 cm) are shown in Supplementary Figs. 3 and 4, and Fig. 2b, showing white, yellow, and purple colors, respectively (Supplementary Fig. 5, Fig. 2i and Supplementary Note 3). Scanning electron microscopy (SEM) micrographs show that lots of ZIF-67

particles are evenly distributed on the carboxymethylated fiber surface, forming a MOF coating (Fig. 2c). The morphology structure of particles exhibits rhombic dodecahedron, which is the classic feature structure of ZIF-67[41] (Supplementary Fig. 6). In marked contrast, the expected MOF coating is not formed on the unmodified fiber surface (Supplementary Fig. 7 and Supplementary Note 4). Energy-dispersive spectroscopy (EDS) was used to characterize the distribution of ZIF-67

on the fiber surface (Supplementary Fig. 8). Only carbon, oxygen, nitrogen, and cobalt four elementals are observed in the EDS spectrum in Supplementary Fig. 8b, which are the primary elements of cotton textiles and ZIF-67 (Supplementary Fig. 8a), suggesting that the MOF coating via the in situ growth would be more uniform and denser on the carboxymethylated cotton fibers via diazonium chemistry modification.

To further study the chemical structure of ZIF-67-CT, the ATR-FTIR and XPS were performed. Compared to the spectrum of cotton textiles, there are C=O (COOH) and C–C (Ar) two absorption peaks appeared in the carboxymethylated cotton textiles, demonstrating the formation of CT-COOH (Fig. 2d). The new peaks of C=N, C–N, and Co–N are the characteristic absorption peaks of ZIF-67, corresponding to the ligand of 2-methylimidazole coordination with $Co^{2+}$ (Supplementary Fig. 9). As observed in Fig. 2e, the results from XPS indicate the presence of carbon, oxygen, nitrogen, and cobalt species in ZIF-67-CT, which are well in agreement with EDS results. The high-resolution Co 2$p$ spectrum of the ZIF-67-CT (Fig. 2f) displays Co 2$p$3/2 at 781.6 eV and Co 2$p$1/2 peaks at 797.3 eV, and apparent satellite peaks at 786.4 and 802.8 eV, respectively, demonstrating the $Co^{2+}$ appeared in ZIF-67-CT. X-ray diffraction (XRD) is used to further confirm the MOF crystal structures. The fabricated ZIF-67-CT and ZIF-67 have the same crystal characteristic peaks, consistent well with the simulated crystal pattern of ZIF-67, whereas only three apparent wide typical cellulose structure peaks appeared in the XRD pattern of cotton textile (Fig. 2g), confirming that the ZIF-67 coating was successfully formed on the carboxymethylated cotton textiles surface via in situ growth. The BET surface area of samples was measured by $N_2$ adsorption–desorption isotherm (Fig. 2h), and the BET surface area of unmodified cotton textile CT is 21.6 m$^2$ g$^{-1}$, whereas the ZIF-67-CT is increased to 244.2 m$^2$ g$^{-1}$ due to the mesoporous structure of ZIF-67 (1750.4 m$^2$ g$^{-1}$, Supplementary Fig. 10), indicating that MOF-textile was successfully fabricated and the MOF coating could significantly increase the surface area of textile. In addition, the MOF coating content was calculated to be ~11.4% according to ICP-OES analysis[42]. These morphologies and crystal structure results, along with the spectroscopic findings and the color changes, demonstrate that ZIF-67 coating is successfully constructed on the carboxymethylated cotton fiber surface via the in situ growth.

## Multifunctional performances

We used 1H,1H,2H,2H-perfluorooctyltriethoxysilane as the hydrophobic material to fabricate superhydrophobic ZIF-67-CT textile by thermal crosslinking approach. The XPS and XRD results demonstrated that the ZIF-67-CT was successfully modified with 1H,1H,2H,2H-perfluorooctyltriethoxysilane, and the crystal structure was well maintained (Supplementary Figs. 11 and 12). Figure 3a, b shows pictures of a static and sliding water droplet on the hydrophobic modified ZIF-67-CT surface. The water contact angle (WCA) and water sliding angle (WSA) are 164.9° and 3.5°, respectively, showing excellent superhydrophobicity (WCA > 150° and WSA < 10°). When the sample was immersed in water, there was an air pocket layer on the superhydrophobic ZIF-67-CT surface (Supplementary Fig. 13). This performance was caused by the micro-nano structures of MOF coating on the textile surface, which would lock lots of air on the interface between the superhydrophobic ZIF-67-CT surface and water. As displayed in Fig. 3c, the water adhesive force on the superhydrophobic ZIF-67-CT surface is as low as 27.64 μN, indicating that the interaction force between the sample surface and water is very weak. As displayed in Supplementary Fig. 14, the modified ZIF-67-CT exhibits excellent superhydrophobic and superoleophilic properties, and shows good acid and alkali resistance stability (Supplementary Note 5, Supplementary Figs. 15 and 16). Owing to its excellent superhydrophobic properties, the ZIF-67-CT cannot be fouled with the contaminant and dyed wastewater, showing excellent self-cleaning and antifouling

performances (Supplementary Figs. 17 and 18). The superhydrophobic ZIF-67-CT can also separate different oil-water mixtures, such as petroleum ether and carbon tetrachloride mixed with water, and the separation efficiency was up to 98.6%. After 15 cycles of separation, the separation efficiency remained above 97%, exhibiting excellent recyclability and durability (Supplementary Fig. 19, Supplementary Note 6, and Supplementary Movie 1).

To investigate the UV resistance of superhydrophobic ZIF-67-CT, we tested its UV transmittance curves on a Labsphere UV-2000 transmittance analyzer. The UV transmittance of ZIF-67-CT and superhydrophobic ZIF-67-CT is significantly lower than CT, indicating that most of UV is blocked by the MOF-textile (Fig. 3d). Consequently, the UVA and UVB of MOF-textile are lower than unmodified cotton textile (Fig. 3e), which may be due to the transition of electrons[43] of the $Co^{2+}$ of ZIF-67. Specifically, when there is no external stimulus light, the electrons of metal in the MOF will stay in the valence band of low energy. Once the ZIF-67 is irradiated by the incident light, it absorbs the UV photonic energy, enabling the transition of electrons from the valence band to the conduction band with high energy. Figure 3f shows that the UPF value of unmodified cotton textile is 25.46 ± 3.16, which is lower than 50, indicating poor UV resistance, whereas the UPF values of ZIF-67-CT and superhydrophobic ZIF-67-CT are higher than 50, showing excellent UV resistance property. This result demonstrates that the ZIF-67-CT textile can be a good UV blocker that protects the skin from UV rays.

We also evaluated the anti-icing ability of superhydrophobic ZIF-67-CT on a Peltier cooler table with an IR camera. One water droplet went through four stages from liquid transform into ice: normal liquid, supercooling liquid, phase transition liquid-solid, and fully frozen solid stages (Fig. 3g). Compared to unmodified cotton textile, the total frozen time of superhydrophobic ZIF-67-CT is delayed from 90.5 s to 223.7 s (Fig. 3h), indicating excellent anti-icing property. The result indicates that the superhydrophobic ZIF-67-CT could delay the icing and freezing processes by reducing the contact area and increasing heat transfer resistance between the water droplet and substrate surface. These results suggest that the superhydrophobic ZIF-67-CT has high application potential in UV resistance, anti-icing, antifouling, self-cleaning, and oil-water separation.

## Antibacterial activity

The antibacterial activity of the ZIF-67-CT and carvacrol-loaded ZIF-67-CT (ZIF-67-CT/Carvacrol) were tested using the disc-diffusion analytic method and plate count method, including E. coli and S. aureus. Figure 3i displays the photographs of LB agar plates with CT, ZIF-67-CT, and ZIF-67-CT/Carvacrol being inoculated with E. coli for 24 h, and others in Fig. 3j with S. aureus being incubated for 24 h. Supplementary Fig. 20 shows that the inhibition zones of unmodified cotton textiles are 0.0 mm, indicating no antibacterial activity against E. coli and S. aureus. For ZIF-67-CT, the inhibition zones are increased to 0.5 and 2.5 mm against E. coli and S. aureus, respectively, showing well antibacterial activity due to the kill bacterial effect of metal ion of MOF[44]. Interestingly, the inhibition zone of ZIF-67-CT/Carvacrol inoculated E. coli is increased to 33.0 mm, especially for S. aureus (90 mm), demonstrating excellent antibacterial activity. This result indicates that the loaded fragrance carvacrol releases the encapsulated molecules continuously into the agar plate inoculated bacteria, benefiting from the large specific surface area and high porosity of ZIF-67[14]. The differences in inhibition zones against E. coli and S. aureus might be caused by their unique biological cell membrane structures[45].

Figure 3k, l shows that LB agar plates were inoculated with E. coli and S. aureus being treated using CT, ZIF-67-CT, and ZIF-67-CT/Carvacrol for 18 h. No colonies appeared on the agar plates inoculated with ZIF-67-CT and ZIF-67-CT/Carvacrol treated E. coli and S. aureus, while a large number of colonies survived on the agar plates inoculated with CT-treated E. coli and S. aureus, indicating that the colony number

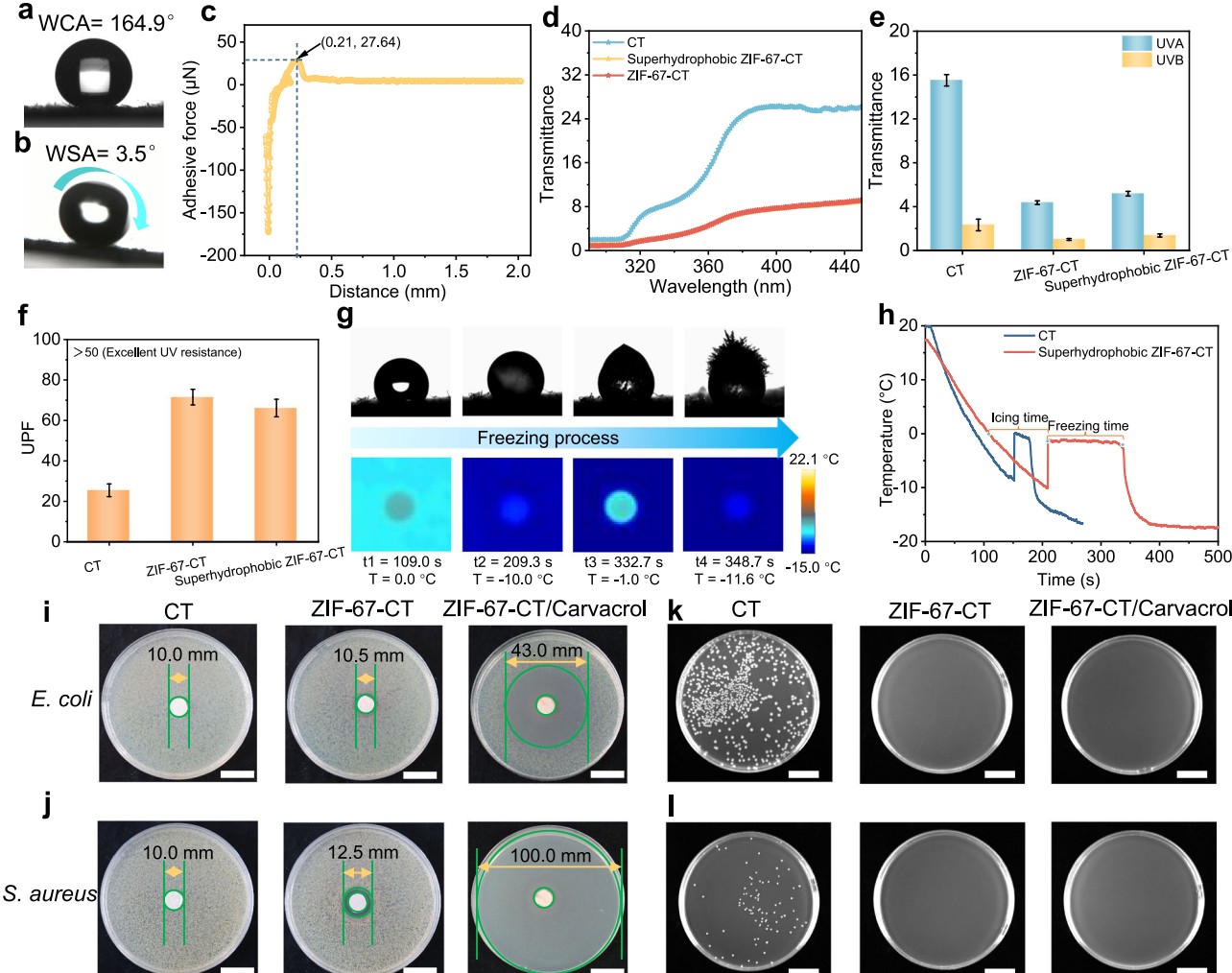

**Fig. 3 | Multifunctional performance of superhydrophobic ZIF-67-CT and the antibacterial activity of CT, ZIF-67-CT, and ZIF-67-CT/Carvacrol. a, b** Pictures of the (**a**) static and (**b**) sliding water droplet on the hydrophobic modified ZIF-67-CT surface. **c** Water adhesive force-distance curve of superhydrophobic ZIF-67-CT. **d–f**, (**d**) UV transmittance curves, (**e**) UVA and UVB, (**f**) UPF values of pristine cotton textile CT, ZIF-67-CT, and superhydrophobic ZIF-67-CT, respectively. **g** Pictures and IR images of water droplets on superhydrophobic ZIF-67-CT surface at different freezing states. **h** Temperature-time curve of the water droplet on CT and superhydrophobic ZIF-67-CT surface during the cooling condition. **i, j** Photographs of the LB agar plates with CT, ZIF-67-CT, and ZIF-67-CT/Carvacrol inoculation with (**i**) *E. coli* and (**j**) *S. aureus* and incubation for 24 h (Disc-diffusion analytic method. The diameter of samples is 10 mm. Scale bars, 20 mm). **k, l** Photographs of LB agar plates after the 18 h inoculation with (**k**) *E. coli* and (**l**) *S. aureus* and incubation of the CT, ZIF-67-CT, and ZIF-67-CT/Carvacrol treated bacteria cultures for 18 h (Plate count method). Scale bars, 20 mm.

for the ZIF-67-CT and ZIF-67-CT/Carvacrol treated bacterial cultures were lower than those for the CT-treated cultures. We calculated the antibacterial efficiency using the plate count method (Supplementary Fig. 21) and found the antibacterial efficiency of ZIF-67-CT and ZIF-67-CT/Carvacrol against *E. coli* and *S. aureus* was 99.99%. These results demonstrate that the ZIF-67-CT exhibits superior antibacterial activity against bacteria, especially the carvacrol-loaded ZIF-67-CT. Analogously, the MOFs-textiles could extend to encapsulate more types of essential oils, which are superior to the conventionally modified textiles as promising antibacterial textiles.

**Degradation ability and mechanical properties**

We evaluated the degradation ability of ZIF-67-CT using various dyes as simulated organic contaminations, and the degradation effect was measured using a UV-vis absorption spectrophotometer. Figure 4a shows the methylene blue (MB) dyed water changed to a clean and colorless solution after the degradation separation process (Supplementary Fig. 22), indicating that the ZIF-67-CT exhibited excellent catalytic performance. We then investigated the influence of the PMS

concentration on the ZIF-67-CT catalytic degradation and the dye degradation kinetic of ZIF-67-CT with PMS at room temperature (Supplementary Note 7, Supplementary Figs. 23, 24, 25 and 26). Figure 4b, c shows that the ZIF-67-CT has a similar degradation effect on RhB and MO solutions as the MB solution. To study the catalytic degradation mechanism of the ZIF-67-CT/PMS, electron paramagnetic resonance (EPR) was used to detect the generated free radicals in the catalytic degradation reaction systems. There were no signal peaks observed in the PMS system, whereas four peaks were found after the addition of ZIF-67-CT (Fig. 4d). This result suggests that radical species $\cdot SO_4^-$ and $\cdot OH$ are generated during the PMS activation, and $\cdot OH$ is contributed dominantly to the degradation of dyes solution. This result demonstrated that the ZIF-67-CT textile exhibited outstanding MB, RhB, and MO degradation ability via PMS activation.

Next, to evaluate the ZIF-67-CT's laundering stability, a piece of ZIF-67-CT textile was washed based on the international standard ISO 105 C10 (Fig. 4e and Supplementary Movie 2), and slight color changes, and no obvious changes of decreased integrity were observed. We also confirmed that the MOF structure was well maintained by

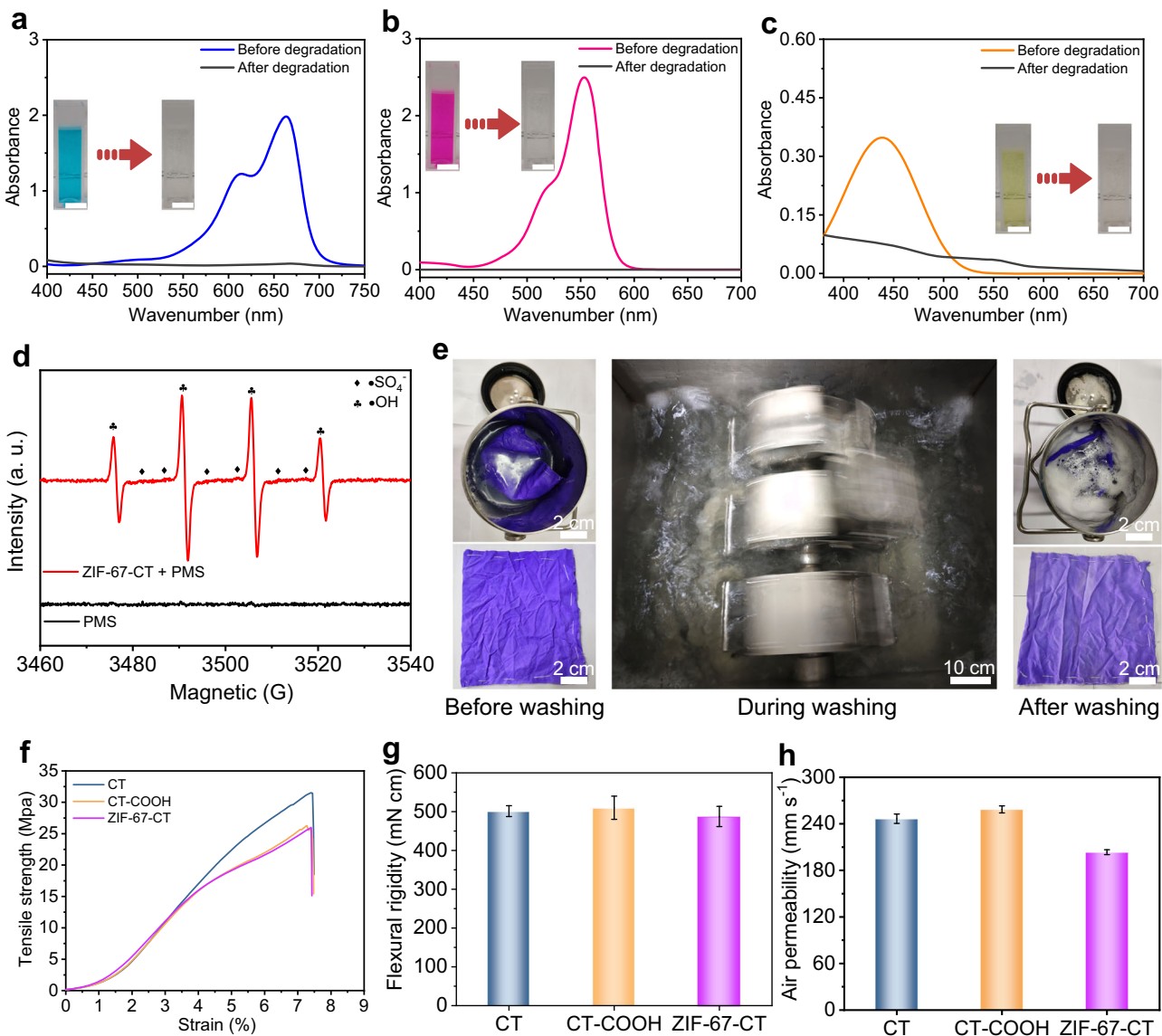

**Fig. 4 | Degradation and physical performance of the ZIF-67-CT. a–c** UV-vis absorption spectra of the (**a**) MB, (**b**) RhB, and (**c**) MO solutions before and after the degradation by ZIF-67-CT. Scale bars, 1 cm. **d** EPR spectra of DMPO as the trapping agent. **e** Photographs of a ZIF-67-CT being washed based on the international standard ISO 105 C10. **f–h** The (**f**) tensile strength curves, (**g**) flexural rigidity, and (**h**) air permeability of CT, CT-COOH, and ZIF-67-CT.

SEM (Supplementary Fig. 27), XRD (Supplementary Fig. 28), and $N_2$ adsorption–desorption isotherm of ZIF-67-CT after laundering (Supplementary Fig. 29). The SEM images of the ZIF-67-CT after washing show that some MOF particles off the fiber surface at some locations, but most MOF coating is well maintained as before washing, and the XRD spectrum shows a similar diffraction pattern and has the same crystal characteristic peaks, demonstrating ZIF-67-CT shows great laundering stability. We also tested the physical mechanical properties of ZIF-67-CT. Figure 4f shows the tensile strength of ZIF-67-CT was 25.98 MPa, ~17% lower than that of the pristine cotton textile CT (31.43 MPa), whereas the elongation was not changed. Besides, the tensile strength of CT-COOH was 26.28 MPa, similar to ZIF-67-CT (25.98 MPa), which is lower than that of the pristine cotton textile CT (31.43 MPa), demonstrating the decrease of the tensile strength of cotton fiber mainly caused by the acid solution treatment during the carboxymethylation of CT in HCl solution[46]. We further tested the flexural rigidity and air permeability performances of the CT, CT-COOH, and ZIF-67-CT samples. The flexural rigidity was slightly decreased from 501.4 mN cm to 487.5 mN cm (Fig. 4g), and air

permeability was decreased from 252.7 mm s$^{-1}$ to 203.5 mm s$^{-1}$ (Fig. 4h), which may be attributed to the loaded MOF coating on the fiber surface of ZIF-67-CT. These results demonstrate no obvious change in physical-mechanical properties, which does not affect daily use.

**Demonstration of scalable manufacturing**
We further demonstrated that the two steps modification process (Fig. 5a) for manufacturing the ZIF-67-CT is scalable using a facile dipping method, including the carboxymethylation of cotton fiber in 3-aminobenzoic acid diazonium salt HCl solution via diazonium chemistry, MOF in situ growth on the carboxymethylated fiber surface in ligand-contenting and metal-contenting methanol solution via facile soaking. In addition, the textiles after each modification needed to be washed and dried, and the methanol solution was recycled and used to wash the ZIF-67-CT textiles. Figure 5b displays an example of the scale-up preparation, where a roll of ZIF-67-CT cloth was fabricated from commercially available cotton cloth, and the textile's color changed from white to yellow and purple as the carboxymethylation and ZIF-67 in situ growth, corresponding to CT, CT-COOH, and ZIF-67-CT. The

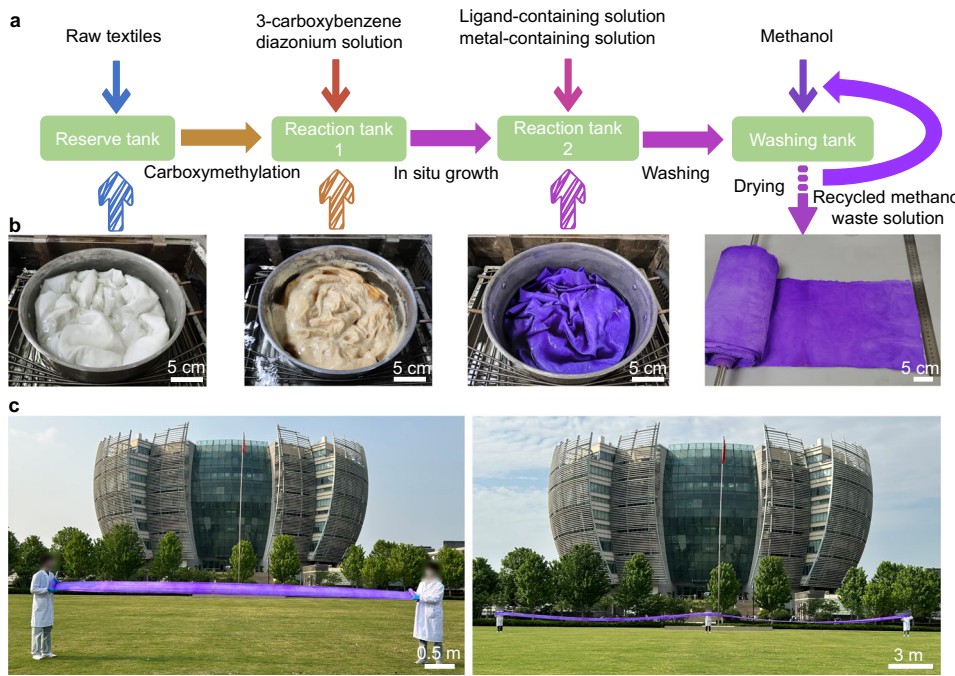

**Fig. 5 | Scalable manufacturing of the ZIF-67-CT. a** The fabrication process of the ZIF-67-CT via in-situ growth. **b** Photographs showing the fabrication process of ZIF-67-CT. **c** Photographs of the large-scale ZIF-67-CT (30 m × 0.25 m).

original cotton cloth was placed in a diameter of 27 cm and a height of 15 cm container filled with 3-aminobenzoic acid diazonium salt HCl solution and soaked for 3 h. After washing and drying, the resultant CT-COOH was soaked in the same container filled with ligand-contenting and metal-contenting methanol solution for 12 h, and after washing and drying, a roll of ZIF-67-CT cloth 0.25 m wide and 30 m long was produced (Fig. 5c and Supplementary Movie 3). It is worth noting that the inherent color of ZIF-67-CT was purple, which could reduce the dyeing process and thus mitigate the environmental impact[47].

Moreover, to demonstrate the feasibility of the facile scalable technology toward other MOFs and cellulose fiber-based materials, we carried out the in situ growth of ZIF-8, UiO-66-NH$_2$, and MOF-303 on carboxymethylated cotton textiles, and the ZIF-67 and ZIF-8 on linen and paper using the same method (Supplementary Note 8). The successful preparation of the MOFs-cellulose fibers was confirmed by SEM (Supplementary Figs. 30, 32, and 34), XRD, and N$_2$ adsorption–desorption isotherms (Supplementary Figs. 31, 33, and 35). The fabricated UiO-66-NH$_2$-CT and MOF-303-CT were measured using SEM and XRD (Supplementary Figs. 36 and 37). These results demonstrate that this facile scalable method is not only used for cotton textile and ZIF-67 MOF but is also applicable to other MOFs and cellulose fiber-based materials.

## Discussion

In this work, we develop a facile, low-energy, and general technology for efficiently fabricating scalable multifunctional MOF-textiles via diazonium chemistry and in situ growth of MOFs on cellulose fibers. Different from existing coating approaches, a MOF coating is anchored by directly binding the metal ions onto the carboxymethylated fibers with carboxyl polymer brushes, thus forming nucleation sites for the growth of MOF crystallites. The ZIF-67-CT shows excellent UV resistance and organic contamination degradation via PMS activation due to the ZIF-67 coating on the fiber surface. The ZIF-67-CT is also used to encapsulate essential oils (for example, carvacrol), and the ZIF-67-CT/Carvacrol exhibits superior antibacterial activity against *E. coli* and *S. aureus*, which is likely attributed to the synergistic sterilization effect of cobalt ions and carvacrol, especially the released of carvacrol. In addition, the superhydrophobic ZIF-67-CT textile is prepared by

directly tethering a hydrophobic molecular layer onto the MOF coating surface, which displays excellent self-cleaning, antifouling, and oil-water separation performances. Besides ZIF-67 MOF and cotton textiles, our developed strategy is also applied to other MOFs (for example, ZIF-8, UiO-66-NH$_2$, and MOF-303) and cellulose fiber-based materials (for example, linen and paper). This general strategy is highly scalable and shows great potential for the mass production of multifunctional MOF-based textiles used in wastewater purification separation membranes, the fragrance industry, and outdoor products.

## Methods

### Preparation of 3-aminobenzoic acid diazonium salt HCl solution
3.3 mmol of NaNO$_2$ (227.68 mg, 99%, Adamas) was dissolved in 60 mL of 1 mol L$^{-1}$ HCl (37 wt%, Qiangsheng) solution, followed by adding 3 mmol of 3-aminobenzoic acid (411.41 mg, 99%, Adamas)[48]. The solution was oscillated for 1 h at 0 °C in the oscillating ice-water bath, forming the 3-aminobenzoic acid diazonium salt HCl solution.

### Preparation of metal-contenting and ligand-contenting solutions
The metal-organic framework was synthesized using metal-contenting and ligand-contenting solutions. Typically, for the MOF ZIF-67, 1.25 mmol of Co(NO$_3$)$_2$ 6H$_2$O (363.79 mg, ≥ 99.95%, Aladdin) was ultrasonically dissolved in 20 mL of methanol (≥ 98%, Qiangsheng), achieving the Co$^{2+}$-contenting solution. Similarly, 12.5 mmol of 2-MeIm (1026.33 mg, ≥ 99.99%, Sigma Aldrich) was ultrasonically dissolved in 30 mL of methanol, achieving the ligand-contenting solution. For the MOF ZIF-8, 1.25 mmol of Zn(NO$_3$)$_2$ 6H$_2$O (371.86 mg, 99%, Sigma Aldrich) was ultrasonically dissolved in 25 mL of methanol, achieving the Zn$^{2+}$-contenting solution. Similarly, 5.05 mmol of 2-MeIm (414.64 mg) was ultrasonically dissolved in 25 mL of methanol, achieving the ligand-contenting solution.

### Preparation of hydrophobic finishing solution
A 5 wt% hydrophobic finishing solution was made by dissolving 1H,1H,2H,2H-perfluorooctyltriethoxysilane (≥ 99.8%, Adamas) in ethanol (95%, Qiangsheng).

## Preparation of MOF-textiles

We fabricated the MOF-textiles through the following two steps. (1) The CT (114 g/m$^{-2}$, Shengbaolu Textile Co., Ltd., China) was soaked in 3-aminobenzoic acid diazonium salt HCl solution, then 53 mg of vitamin C (AR, Sigma Aldrich) was added into the above reactive solution and incubated for 12 hours at room temperature. The textile was thoroughly washed with water until the ungrafted chemicals were completely removed, then dried at room temperature, with the product referred to as CT-COOH. (2) The CT-COOH was soaked in the Co$^{2+}$-contenting methanol solution for 1 hour, then the 2-MeIm-contenting methanol solution was slowly added and incubated for 12 hours at 30 °C. The textile was washed with methanol several times until the nonbonded MOF particles and chemicals were completely removed and were then dried, with the product referred to as ZIF-67-CT. For the preparation of ZIF-67-Linen and ZIF-67-Paper, the linen and paper were treated using the same steps. However, for the preparation of ZIF-8-CT, ZIF-8-Linen, and ZIF-8-Paper were treated using the same steps as ZIF-67-CT, except that the material was soaked in the Zn$^{2+}$-contenting methanol solution, instead of the Co$^{2+}$-contenting methanol solution in step 2.

## Scale-up manufacturing of ZIF-67-CT

(1) One roll of cotton textile (30 m × 0.25 m) was soaked in 4 L of 3-aminobenzoic acid diazonium salt HCl solution, then 3.5 g of vitamin C was added into the above reactive solution and incubated for 3 hours at room temperature. The textile was thoroughly washed with water and dried at room temperature. (2) The CT-COOH was soaked in the 2 L of Co$^{2+}$-contenting methanol solution (Co(NO$_3$)$_2$·6H$_2$O, 24 g) for 1 hour, then the 2 L of 2-MeIm-contenting methanol solution (68.9 g) was slowly added and incubated for 12 hours at 30 °C. The textile was washed with methanol several times until the nonbonded MOF particles and chemicals were completely removed and were then dried, with the product referred to as ZIF-67-CT (30 m × 0.25 m).

## Preparation of UIO-66-NH$_2$-CT

ZrCl$_4$ (1.258 g, 99.9 %, Shanghai Aladdin) was dissolved into a mixture of N,N-dimethylformamide (DMF, 50 mL, analytically pure, Qiangsheng), and HCl (37 wt%, 10 mL), obtaining a Zr$^{4+}$-contenting solution. H$_2$N − H$_2$BDC (1.358 g, 98 %, Anhui Zesheng Technology Co., Ltd.) was dissolved into 100 mL of DMF in a 250 mL round-bottom flask, obtaining a ligand-contenting solution. Then, one piece of CT-COOH textile (6 cm×6 cm) was soaked in the Zr$^{4+}$-contenting solution for 10 min, then the H$_2$N − H$_2$BDC-contenting solution was slowly added and incubated for 4 hours at 80 °C. Finally, the prepared textile was washed with DMF and ethanol several times, respectively, dried at 37 °C, and activated at 80 °C under vacuum, with the product referred to as UIO-66-NH$_2$-CT.

## Preparation of MOF-303-CT

AlCl$_3$·6H$_2$O (3.60 g, 99%, Sigma Aldrich) was dissolved into 75 mL of DI water, obtaining an Al$^{3+}$-contenting solution. H$_2$PZDC·H$_2$O (2.61 g, 97%, Sigma Aldrich) was added to a 75 mL NaOH solution (12 g L$^{-1}$, 97%, Sigma Aldrich), obtaining a ligand-contenting solution. One piece of CT-COOH textile (6 cm × 6 cm) was soaked in the Al$^{3+}$-contenting solution for 30 min, then the textile was transferred to the ligand-contenting solution. Then, the Al$^{3+}$-contenting solution was slowly added, and reflux reacted for 3 hours at 120 °C. Finally, the prepared textile was washed with water and methanol several times, respectively, dried at room temperature, and activated at 80 °C under vacuum, with the product referred to as MOF-303-CT.

## MOFs loading test

The normalized BET surface area calculation and ICP-OES/MS methods were used to test the MOF coating content of ZIF-67-CT. The ZIF-67 loading on fiber was calculated to be 11.4% based on ICP-OES analysis

(Thermo Scientific iCAP PRO), which is similar to the result calculated based on BET surface area calculation (12.7%). The MOF mass loading (MML) is calculated using the following equation:

$$MML\,(\%) = \frac{C_1 \times V_0}{m_0 \times W_{Co} \times 10^6} \times 100\% \qquad (1)$$

where $C_1$ is the concentration of Co in the diluted nitric acid solution measured by ICP-OES in mg L$^{-1}$, $V_0$ is the volume of the test solution in mL, $m_0$ (g) is the quality of the test sample, and $W_{Co}$ is the mass percentage in MOF.

## Preparation of carvacrol-loaded ZIF-67-CT

The essential oil carvacrol (≥ 97%, MERYER) was encapsulated into ZIF-67-CT by one simple soaking method[14]. A piece of prepared ZIF-67-CT was immersed in carvacrol and oscillated in the shaker for 10 hours. The sample was taken out and washed with hexane, then dried for 2 hours at 37 °C.

## Carvacrol loading

The carvacrol loading content was measured by the gravimetric method. Owing to the large surface area and porosity of ZIF-67, the carvacrol loading content of ZIF-67-CT was up to ~ 90 mg g$^{-1}$, whereas the CT was only about 3 mg g$^{-1}$.

## Preparation of superhydrophobic MOF-textiles

The superhydrophobic MOF-textiles were fabricated by a thermal crosslinking approach. Briefly, the ZIF-67-CT was immersed in the hydrophobic finishing solution with 5 wt% of 1H,1H,2H,2H-perfluorooctyltriethoxysilane for 1 hour at 30 °C, then baked at 130 °C for 2 hours, obtaining the superhydrophobic ZIF-67-CT.

## Characterization

The morphologies and elemental distribution of samples were observed using a Hitachi S-8100 field emission SEM coupled with an energy-dispersive X-ray spectroscopy system and a Hitachi HT7700 transmission electron microscopy (TEM). The N$_2$ adsorption–desorption isotherms were performed using a Micromeritics TriStar II 3020 analyzer, and the corresponding specific surface areas were calculated by the BET method. The chemical and crystal structures of prepared samples were characterized by ATR-FTIR spectroscopy (Nicolet 5700, Thermo Electron), XPS (Thermo Scientific KAlpha), and XRD (D8 Advance, Bruker).

The wettability of samples was conducted on a DSA100 Krüss instrument, and the WCA and WSA result values of every sample were taken as an average of five results. In addition, the water adhesive force-distance curve was tested using a Dataphysics DCAT 11 instrument.

The tensile and elongation properties of CT, CT-COOH, and ZIF-67-CT were characterized using an INSTRON-5967 mechanical tester. The tested textiles were 5 cm wide and 25 cm long, and were uniaxially stretched at a strain rate of 0.5 cm min$^{-1}$. The air permeability (sample size, 20 cm × 20 cm) and flexural rigidity (sample size, 20 cm × 5 cm) were measured to determine the physics performances of samples based on international standards ISO 9237/7231 and ISO 9073-7 using a YG461G fully automatic air permeability tester and a YG (B) 022D automatic fabric stiffness tester, respectively. The results for each textile were evaluated from the average value of three samples.

The laundering procedure used to test the durability of ZIF-67-CT was based on the international standard ISO 105 C10. An SW20A-IIA washing machine was used. A piece of ZIF-67-CT sample was immersed into the washing tank with 4 g L$^{-1}$ of standard detergent (200 mL). The laundering procedure was processed for 30 min at 40 °C with a rotating speed of 40 rpm.

A Bruker EMX PLUS EPR was used to detect the generated free radicals in the catalytic degradation reaction systems.

### UV resistance assessment

The UV-resistance performance of CT, ZIF-67-CT, and super-hydrophobic ZIF-67-CT was performed on a Labsphere UV-2000 transmittance analyzer, and the UPF, UVA, and UVB data were recorded to characterize the UV-resistance ability of textiles. The results for each textile were evaluated from the average value of three samples.

### Anti-icing assessment

The CT and superhydrophobic ZIF-67-CT were fixed on a Peltier cooler table with an IR camera. One 5 μL of water droplet was placed on the sample surface, and the cooler table was cooled from 20 °C to − 20 °C with a cooling rate of 5 °C min$^{-1}$. The real-time temperature of the water droplet was recorded by the IR camera.

### Oil-water separation test

To assess the oil-water separation ability of superhydrophobic ZIF-67-CT, the oil-water solution was prepared by mixing with the oil phase (dyed red) and water phase (dyed blue). Carbon tetrachloride (CCl$_4$, 99%, Qiangsheng) and petroleum ether (AR, Qiangsheng) were chosen as the heavy oil phase and light oil phase, respectively. The separation efficiency $\eta$ (%) was calculated according to the equation:

$$\eta\,(\%) = Vc/Vo \times 100\% \qquad (2)$$

where $VO$ and $Vc$ represented the volumes of collected water before and after separation.

### Antibacterial assessment

For the antibacterial assessment assay, the prepared textiles with 10 mm in diameter were used and sterilized under an ultraviolet lamp for 30 min. The disc-diffusion analytic method and plate count method were used to evaluate the antibacterial activity of prepared textile samples against *E. coli* ATCC25922 and *S. aureus* ATCC29213, in which the seed cultures were prepared in LB media (A507002, Sangon Biotech Co., Ltd.) for 15 hours at 37 °C and 200 rpm shaking. The prepared seed bacterial cultures were then diluted to 10$^6$ CFU mL$^{-1}$ with sterile PBS solution.

For the disc-diffusion analytic method, 100 μL of diluted bacterial cultures were evenly spread on LB solid medium along with the sterilized textile sample, subsequently incubated for 24 hours at 37 °C, and then the observed inhibition zone of the samples was recorded using a camera. The inhibition zone diameter was calculated according to the formula:

$$d = da - 10\,(mm) \qquad (3)$$

where $da$ was the diameter of the LB agar plate with the sample, and 10 mm was the diameter of the samples.

For the plate count method, 2 mL of diluted bacterial cultures were incubated with the sterilized textile sample in LB solid medium for 18 hours at 37 °C and 150 rpm shaking. The bacteria cultures were serially diluted 10-fold, and then 100 μL of diluted bacterial cultures were incubated again for 18 hours at 37 °C in an LB solid medium. Finally, the plates were imaged, and the number of living colonies was counted by the agar plate count method.

### Catalytic degradation performance assessment

To assess the degradation performance of ZIF-67-CT textiles, we tested the ZIF-67-CT textiles with 50 mg L$^{-1}$ of dyes water solutions MB (≥ 98%), RhB (AR), and MO (≥98%, Shanghai Macklin Biochemical Co., Ltd.) at 25 °C, and the initial pH value of solutions was 6.8. Briefly, a certain amount of PMS (> 4.0% (active oxygen basis (by Na$_2$S$_2$O$_3$, titration)), Sigma Aldrich) was added to the dye solution and completely dissolved by stirring for 10 min. Then, the multilayer ZIF-67-CT textile membrane was placed in a conical glass funnel, designing a

simple device. Subsequently, the prepared solution was continuously dripped and filtrated from the textiles. The absorbance value of dye solutions before and after degradation by ZIF-67-CT textiles was measured using a UV-vis absorption spectrophotometer. The organic contaminants degradation efficiency $\rho$ (%) was calculated following the formula:

$$\rho\,(\%) = \left(1 - \frac{Ca}{Cb}\right) \times 100\% \qquad (4)$$

where $Cb$ and $Ca$ are the absorbances of the solutions before and after degradation.

### Reporting summary

Further information on research design is available in the Nature Portfolio Reporting Summary linked to this article.

## Data availability

The data that supports the findings of the study are included in the main text and supplementary information files. Raw data can be obtained from the corresponding author upon request.

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

## Acknowledgements

We acknowledge the support from the National Natural Science Foundation of China (51673137, Z.L.), China Scholarship Council (CSC, 202106920042, W.L.), and the joint scientific research project of the Sino-foreign cooperative education platform of Jiangsu Higher Education Institutions (5011500720, Z.L.). This work was supported by the Singapore Ministry of Education Academic Research Fund Tier 2 (MOE2019-T2-2-127, MOE-T2EP50120-0002, and MOE-T2EP50123-0014, L.W.), the Singapore Ministry of Education Academic Research Fund Tier 1 (RG62/22, L.W.), A*STAR under AME IRG (A2083c0062, L.W.), A*STAR under IAF-ICP Programme I2001E0067 and the Schaeffler Hub for Advanced Research at NTU (L.W.), the IDMxS (Institute for Digital Molecular Analytics and Science) by the Singapore Ministry of Education under the Research Centers of Excellence scheme, and the NTU-PSL Joint Lab collaboration (L.W.). We also thank Jialiang Kang for the showing process of the large-scale ZIF-67-CT in Fig. 5c and Supplementary Movie 3.

## Author contributions

W.L., Z.L., S.C.T., and L.W. conceived the idea and supervised the experiments. W.L. performed sample fabrication, structures, and morphologies characterization. Z.Y. and Y.Z. performed the ATR-FTIR, XPS, and catalytic degradation performance measurements. C.L. and X.H. performed the TEM measurement and the showing process of the

large-scale ZIF-67-CT (30 m × 0.25 m). S.W., Z.W., B.H., and S.Y. performed the XRD and SEM measurements. J.X., Y.L., and T.Z. performed the ultraviolet resistance assessment and tensile properties measurements. W.L., Z.L., S.C.T., and L.W. wrote and revised the manuscript. All authors discussed the results and implications and commented on the manuscript at all stages.

## Competing interests

The authors declare no competing interests.
