## [Peer Review File · Nature Communications]

Scalable multifunctional MOFs-textiles via diazonium chemistryReviewers' comments:

Reviewer #1 (Remarks to the Author):

Li et al report developed a scabble synthesis ZIF-67 coated cotton textile. The synthesis and performance are detailed studied. It is a good sample for the field of MOF functionalized composite and the applications are interesting. It could be published after fixing the concern.

Other comments:

1. Several other papers about ZIF-67 coating on textiles should be cited and compared. The current method shows advancement compared with other ZIF-67 coatings, so getting other ZIF-67 coating work involved in the performance factors study (Figure 1 b) is suggested.

2. The sample after washing seems slightly faded, so the residual MOF content after washing testing should be numerically reported. The authors stated that washed ZIF-67 MOF coating structure was well maintained by SEM (Supplementary Fig. S24) and XRD (Supplementary Fig. S25), but the samples are marked as ZIF-8 in captions. The porosity (by N₂ sorption) of the washed sample should be added.

3. The ZIF-67 coating content on cotton seems not correct from the TGA analysis. The residual after calcining is oxides, not ZIF-67. The wrong method will underestimate the MOF content. If the ZIF-69 content is 3.92%, the BET surface area increase from coating component contribution is 5600m²/g. Obviously, it is impossible for ZIF-67. A loading study by ICP-OES test and normalized BET surface area calculation are suggested to understand the coating quality (doi:10.1021/accounts.mr.2c00200)

4. The EPR signal for OH radicals is too small to be visible. It may not be the dominant contributor to dye degradation. The continuous degradation of dye by ZIF-67 is remarkable (Fig. S21). A related work about using MOF/fiber as the catalyst for organic compound continuous degradation by the Farha group could be cited for comparison (doi:10.1002/adma.202300951)

5. ZIF-67-CT and ZIF-67-CT/Carvacrol seems decolorized from purple in the inoculation with (i) *E. coli* and (j) *S. aureus* and incubation for 24 h. Could the authors explain the reason? Carvacrol loading in the composite should be studied.

Reviewer #3 (Remarks to the Author):

In this study, the authors report on a facile and scalable approach for directly integrating ZIF-67 (mainly) and ZIF-8 onto cotton fibers subjected to grafting through diazonium chemistry. The ZIF-67-Cotton textile (ZIF-67-CT) shows various features, including ultraviolet (UV) resistance characteristics, contamination degradation (via peroxymonosulfate activation), self-cleaning, antifouling, and oil-water

separation. However, the reviewer believes that the novelty of this work is absent, including but not limited to preparing carboxymethylated cotton fabric to facilitate an in situ growth of metal-organic frameworks (MOFs) on them, which was already demonstrated in the previous report (Li et al., *ACS Nano* 2022, 16, 14779–14791) from the same research group. The only difference between the previous report (Li et al., *ACS Nano* 2022, 16, 14779–14791) and this manuscript in preparing the composite material is whether metal-containing and ligand-containing precursor solutions are separated. Furthermore, such methods for integrating MOFs into fiber substrates have already been extensively studied. Assessing the novelty of the research and the integrity of the experimental results, I, as a reviewer, do NOT recommend that this paper be published in Nature Communications.

Herein, I put several suggestions and questions regarding the manuscript. I kindly request that the authors address these points and consider submitting this manuscript to other journals.

1. Please add the page number to the manuscript.
2. (Line 50-54 on page 2) Relevant references are absent concerning dip-coating, hot-pressing, and spraying methods for MOF integration, although those are stated in the text.
3. (Line 54-57 on page 2) It seems that the authors claim that the references (#30-#34) only demonstrate the physical incorporation of MOFs into polymeric fiber substrates, but it is not... The key to some of the work the authors of this work refer to is associated with the chemical integration of MOFs into textiles. Please double-check the references and replace them (not related) with relevant ones.
4. (on page 3) It is stated that “In addition, the fabrication of ZIF-67-CT is facile and scalable...” According to the preparation methods of MOF-textiles in this work, it takes more than a day, including a carboxymethylation of cotton fabric and MOF growth onto it. A more detailed explanation regarding the extent of being “facile and scalable” compared to other reported MOFs-textiles is necessary to convince the reviewers and readers.
5. (Line 99-101 on page 4) It is stated that “Scanning electron microscopy (SEM) micrographs show that lots of ZIF-67 particles are evenly distributed on the carboxymethylated fiber surface, forming a MOF coating (Fig. 2c).” The resolution of the SEM image (Fig. 2c, right) is not good enough to tell the morphology of ZIF-67 on the fiber surface. Taking a higher magnification image of the sample is recommended. In addition, Fig. S6 and S7 do not agree with the text. Please revise it.
6. (Line 116-117 on page 4) The authors claim that “as observed in Fig. 2e, the results from XPS indicate the presence of carbon, oxygen, nitrogen and cobalt species in ZIF-67-CT, which are well in agreement with EDS results.” In this work, the authors have only considered the presence of the major elements comprising ZIF-67. However, it is recommended that the authors maximize the utilization of the obtained XPS data by calculating element area ratios from XPS for comparison with those of ZIF-67 synthesized in the liquid phase.
7. (Line 121-125 on page 5) The authors claim that “The fabricated ZIF-67-CT 121 and ZIF-67 have the same crystal characteristic peaks, consistent well with the simulated crystal pattern of ZIF-67...” As shown in Fig. 2g (on page 6), It appears that there are shifts in the major peaks corresponding to ZIF-67 among ZIF-67-CT, ZIF-67 powder, and simulated ZIF-67. In addition, it is highly recommended that the

authors calculate the specific values of the lattice parameters, peak broadness, and peak intensity ratios of the ZIF-67 component on the fabric to investigate any structural differences compared to free-standing ZIF-67 crystal powder.

8. (Line 157-159 on page 7) The authors state that “the modified ZIF-67-CT exhibits excellent superhydrophobic and superoleophilic properties and shows good acid and alkali resistance stability (Supplementary Fig. S15).” Time-dependent chemical stability data for modified ZIF-67-CT should be conducted and included in this work to claim the point above.

9. (Line 170-172 on page 7) The authors mention that “...which may be due to the transition of electrons[36] of the Co^{2+} of ZIF-67.” More detailed but concise information regarding the speculation is required to be included.

10. (Line 221-223 on page 9) The authors claim that “... the ZIF-67-CT exhibits superior antibacterial activity against bacteria...” I understand that the ZIF-67-CT shows antibacterial activity through the data provided in the manuscript. Still, as a reader, I cannot tell how effective the ZIF-67-CT is in comparison with other MOF-textile samples and related materials. Please make a good comparison with other materials in the antibacterial activity.

11. (Line 230-232 on page 9) The authors claim that “... the ZIF-67-CT exhibited excellent catalytic performance.” I cannot find any dye degradation kinetic data (dye concentration vs. time) in the main text as well as in the supplementary document. Please provide the kinetic data.

12. (Line 245-247 on page 10) The authors claim that “The SEM images of the ZIF-67-CT after washing show the same morphology structure as before washing...” However, there appears to be a noticeable change before and after the fabric composite washing (Fig. 4e). It is recommended that the authors take SEM images of the washed samples at varied locations. I do not think the SEM images in Fig. S24 represent the samples after washing.

13. (Line 250-252 on page 10) The authors state that “... the fiber structure damage of ZIF-67-CT during the modification processes...” Here, the authors should characterize CT-COOH and ZIF-67-CT after washing to confirm what causes the fiber structure damage of ZIF-67-CT, and to identify the relationship between fiber treatment methods and fiber mechanical properties.

14. (Line 255-257 on page 10) The authors claim that “These results demonstrate no obvious change in physical-mechanical properties, which does not affect its daily use.” In order to claim that point, the authors should show N_2 isotherms of ZIF-67-CT after washing and compare with those of as-prepared ZIF-67-CT. Surface area, pore volume, and porosity are closely related to MOF-textiles properties, so maintaining such properties after laundering stability tests is critical in this work.

15. (Line 307-309 on page 13) The authors conclude that “Besides ZIF-67 MOF and cotton textiles, our developed strategy can also be applied to other MOFs (for example, ZIF-8 MOF)...” In my humble opinion, demonstrating ZIF-8 integration is not enough to claim the method is generic. The authors need to show other MOFs beyond the ZIFs with SOD (sodalite-like) structures.

Point-by-Point Responses to Referees' Comments

We thank all the referees for their in-depth review of our manuscript and for enriching us with their valuable comments and suggestions, which have helped us further improve the quality of the manuscript. The referees' comments are listed in *blue font color*, and the authors' responses are listed in black font color. All the changes in the revised manuscript and supplementary information are marked in *red font color*.

Referee #1:

Li et al report developed a scabble synthesis ZIF-67 coated cotton textile. The synthesis and performance are detailed studied. It is a good sample for the field of MOF functionalized composite and the applications are interesting. It could be published after fixing the concern.

Response:

We really appreciate your in-depth review of our manuscript. According to your comments and suggestions, we have revised and improved our manuscript.

Other comments:

1. Several other papers about ZIF-67 coating on textiles should be cited and compared. The current method shows advancement compared with other ZIF-67 coatings, so getting other ZIF-67 coating work involved in the performance factors study (Figure 1 b) is suggested.

Response:

We thank the referee for the valuable comment. As shown in Fig. 1b (revised), we have added and compared with other ZIF-67 coating works regarding various key performance factors: scalability, mechanical stability, porosity, fragrance encapsulation, pollutant degradation, and antibacterial activity. As a result, our method shows great advancement in scalability, fragrance encapsulation, and antibacterial activity (Supplementary Note 1). These references about other ZIF-67 coating works have been included in the revised manuscript as refs [39] and [40].

Fig. 1b (revised) Comparison of the performance factors of ZIF-67-CT to those of pristine, conventionally modified cotton textiles, and other ZIF-67 coating textiles.

39. Zhang S, Zhao M, Li H, Hou C, Du M. Facile in situ synthesis of ZIF-67/cellulose hybrid membrane for activating peroxymonosulfate to degrade organic contaminants. *Cellulose* **28**, 3585-3598 (2021).
40. Qiao X, Gao W, Liu X, Fang K, Li Q, Lu X, et al. Preparation of zeolitic imidazolate framework-67/wool fabric and its adsorption capacity for reactive dyes. *Journal of Environmental Management* **321**, 115972 (2022).

2. The sample after washing seems slightly faded, so the residual MOF content after washing testing should be numerically reported. The authors stated that washed ZIF-67 MOF coating structure was well maintained by SEM (Supplementary Fig. S24) and XRD (Supplementary Fig. S25), but the samples are marked as ZIF-8 in captions. The porosity (by N₂ sorption) of the washed sample should be added.

Response:

We thank the referee for the valuable comment. We are very sorry for our mistake about the wrong captions of ZIF-8, which have been corrected to be ZIF-67-CT in Supplementary Fig. S27 (SEM) and Supplementary Fig. S28 (XRD). In addition, we have added the N₂ adsorption–desorption isotherm BET of the washed sample in Supplementary Fig. S29. The BET surface area is 207.6 m² g⁻¹, and the pore width of the mesoporous structure is mainly at ≈ 0.64 nm. The reason for the BET decrease of the washed sample is mainly due to the loss of MOF during the laundering process. Hence, the residual MOF content after the washing testing is calculated to be ~ 10.6% based on the changes in BET surface area (the MOF content before the washing testing is ~ 12.7%).

Supplementary Fig. S29 (a) N₂ adsorption–desorption isotherm and (b) pore width distribution of ZIF-67-CT after the laundering process.

3. The ZIF-67 coating content on cotton seems not correct from the TGA analysis. The residual after calcining is oxides, not ZIF-67. The wrong method will underestimate the MOF content. If the ZIF-69 content is 3.92%, the BET surface area increase from coating component contribution is 5600m²/g. Obviously, it is impossible for ZIF-67. A loading study by ICP-OES test and normalized BET surface area calculation are suggested to understand the coating quality (doi:10.1021/accounts.mr.2c00200)

Response:

We thank the referee for the valuable comment. As suggested by the referee, we have added the normalized BET surface area calculation to test the MOF coating content of ZIF-67-CT. As shown in Fig. 2h and Supplementary Fig. S10, the BET surface area of CT, ZIF-67-CT, and ZIF-67 particles is 21.6, 244.2, and 1750.4 m² g⁻¹, respectively. So, the ZIF-67 coating content of ZIF-67-CT is measured to be 12.7%, according to the normalized BET surface area calculation [45].

We also thank the referee for providing this valuable reference for us to understand the coating quality. This reference has been included in the revised manuscript as ref [45].

Fig. 2h N₂ adsorption-desorption isotherms of CT and ZIF-67-CT.

Supplementary Fig. S10 N₂ adsorption-desorption isotherms of ZIF-67.

4. The EPR signal for OH radicals is too small to be visible. It may not be the dominant contributor to dye degradation. The continuous degradation of dye by ZIF-67 is remarkable (Fig. S21). A

related work about using MOF/fiber as the catalyst for organic compound continuous degradation by the Farha group could be cited for comparison (doi:10.1002/adma.202300951)

Response:

We are really grateful for this comment. We have rechecked the EPR test results of ZIF-67-CT and corrected the signal marked of OH and SO₄⁻ radicals in the revised Fig. 4d. As mentioned by the referee, the signal of OH radicals is significantly stronger than SO₄⁻ radicals. Therefore, the SO₄⁻ radical is not the dominant contributor to the dye degradation, but the OH radical is. The related radical degradation result can be supported by the following refs [R1] and [R2].

R1. Wang F X, Zhang Z C, Yi X H, et al. A micron-sized Co-MOF sheet to activate peroxymonosulfate for efficient organic pollutant degradation. *CrystEngComm* **24**, 5557-5561 (2022).

R2. Zhang S, Zhao M, Li H, et al. Facile in situ synthesis of ZIF-67/cellulose hybrid membrane for activating peroxymonosulfate to degrade organic contaminants. *Cellulose* **28**, 3585-3598 (2021).

Fig. 4d (revised) EPR spectra of DMPO as the trapping agent.

In addition, as one comparison reference, the published work (10.1002/adma.202300951) has been cited in the revised manuscript as ref [11].

11. Ma K, Cheung YH, Kirlikovali KO, Xie H, Idrees KB, Wang X, et al. Fibrous Zr-MOF Nanozyme Aerogels with Macro-Nanoporous Structure for Enhanced Catalytic Hydrolysis of Organophosphate Toxins. *Adv Mater*, 2300951 (2023).

5. ZIF-67-CT and ZIF-67-CT/Carvacrol seems decolorized from purple in the inoculation with (i) *E. coli* and (j) *S. aureus* and incubation for 24 h. Could the authors explain the reason? Carvacrol loading in the composite should be studied.

Response:

We thank the referee for the valuable comment. Indeed, ZIF-67-CT and ZIF-67-CT/Carvacrol are decolorized in the inoculation with (i) *E. coli* and (j) *S. aureus* and incubation for 24 h. However, ZIF-67-CT and ZIF-67-CT/Carvacrol would not be decolorized in the air for a long time (for example, 180 days). Therefore, this phenomenon is mainly due to the fact that Co ions of ZIF-67 would gradually release when incubated in an LB solid medium with bacterial suspension. Importantly, the released Co ions would contact and interact with the proteins and nucleic acids of the bacterial membrane, damaging the bacterial membrane and cell wall, and then leading to bacterial destruction and death, which is the dominant reason for ZIF-67-CT to enable antibacterial activity. This phenomenon and antibacterial performance can be further explained by ref [R3].

As suggested by the referee, we have added the Carvacrol loading in the composite calculated by the gravimetric method. Owing to the large surface area and porosity of ZIF-67, the Carvacrol loading content of ZIF-67-CT was up to about 90 mg g⁻¹, whereas the pristine CT was only about 3 mg g⁻¹.

- R3. Wyszogrodzka G, Marszałek B, Gil B, et al. Metal-organic frameworks: mechanisms of antibacterial action and potential applications. *Drug Discovery Today* **21**, 1009-1018 (2016).

Referee #3:

In this study, the authors report on a facile and scalable approach for directly integrating ZIF-67 (mainly) and ZIF-8 onto cotton fibers subjected to grafting through diazonium chemistry. The ZIF-67-Cotton textile (ZIF-67-CT) shows various features, including ultraviolet (UV) resistance characteristics, contamination degradation (via peroxymonosulfate activation), self-cleaning, antifouling, and oil-water separation. However, the reviewer believes that the novelty of this work is absent, including but not limited to preparing carboxymethylated cotton fabric to facilitate an in situ growth of metal-organic frameworks (MOFs) on them, which was already demonstrated in the previous report (Li et al., ACS Nano 2022, 16, 14779–14791) from the same research group. The only difference between the previous report (Li et al., ACS Nano 2022, 16, 14779–14791) and this manuscript in preparing the composite material is whether metal-containing and ligand-containing precursor solutions are separated. Furthermore, such methods for integrating MOFs into fiber substrates have already been extensively studied. Assessing the novelty of the research and the integrity of the experimental results, I, as a reviewer, do NOT recommend that this paper be published in Nature Communications. Herein, I put several suggestions and questions regarding the manuscript. I kindly request that the authors address these points and consider submitting this manuscript to other journals.

Response:

We appreciate the efforts and time that Referee #2 has devoted to assess our manuscript and provide very valuable and insightful comments. According to your comments and suggestions, we have revised and improved our manuscript.

Firstly, it appears that our manuscript's main dimensions of novelty were not well presented and fully identified. Therefore, we would like to use this response to re-highlight our novelty and share our thoughts about those points that might need our clarification. In our work, we demonstrate a generic strategy for preparing **highly stable, large-scale, multifunctional MOFs-textiles via diazonium chemistry**. The two facile steps are readily accomplished via diazonium chemistry and in situ growth successively by soaking a cotton textile in 3-aminobenzoic acid diazonium salt HCl solution and metal-containing and ligand-containing solution, respectively. Specifically, **the covalent grafted carboxyl polymer chain brushes on the fiber surface create abundant carboxyl group sites, which assist the initial coordination of cobalt ions to the fiber surface at the molecular level** and facilitate the subsequent in situ growth of MOF nanoparticles, **forming a uniform and dense MOFs coating**. This strategy is expected to provide a new avenue for fabricating scalable multifunctional MOFs-textiles.

Secondly, regarding the comment “preparing carboxymethylated cotton fabric to facilitate an in situ growth of metal-organic frameworks (MOFs) on them, which was already demonstrated in the previous report (Li et al., ACS Nano 2022, 16, 14779–14791) from the same research group. The only difference between the previous report (Li et al., ACS Nano 2022, 16, 14779–14791) and this manuscript in preparing the composite material is whether metal-containing and ligand-containing precursor solutions are separated.” We would like to make the following clarifications: In the previous report (Li et al., ACS Nano 2022, 16, 14779–14791, which was included in this

manuscript as ref [41]), the main research contents are focused on the morphology changes with the reaction conditions (Layer-by-Layer cycles and water content percentage), the water stability of CuBTC coating, anti-icing performance and its anti-icing mechanism with COMSOL simulation. However, **our work is unique and more in-depth, as it offers a generic strategy for preparing highly stable, large-scale, multifunctional MOFs-textiles, which is applicable to various types of MOFs** (for example, ZIF-8, UiO-66-NH₂, and MOF-303) **and different cellulose-based fibers** (for example, linen and paper).

Thirdly, regarding the comment “Furthermore, such methods for integrating MOFs into fiber substrates have already been extensively studied.” Indeed, the field of metal-organic frameworks (MOFs) has seen tremendous growth in the past two decades, with extensive research focused on tuning MOF chemistry and pore structures for specific applications [R4-R6]. Most studies have examined MOFs in their powder form, while reports of the integration of MOFs into alternative forms are much fewer. MOFs with large specific surface area, tunable pore size, and excellent thermal and chemical stability, have been considerably applied to fabricate MOF/cellulose fiber composites [R7]. And we agree with the referee that several methods for integrating MOFs into fiber substrates have already been studied in recent years [R8-R10]. However, to the best of our knowledge, **there are no reported methods that could be used to fabricate scalable MOF-Textiles** (for example, tens meters long or larger than 7.5 m² as reported in our work), **or generic and applicable to many types of MOFs** (for example, ZIF-67, ZIF-8, UiO-66-NH₂, and MOF-303). More importantly, in our work, **the fabricated ZIF-67-CT endows the traditional textiles with more unique functions**, which would broaden application prospects in the textile industry to improve the high added value of textiles, such as fragrance encapsulation, antibacterial, anti-fouling, oil-water separation, and wastewater purification.

Lastly, we would like to thank the referee for the valuable and insightful comments on our demonstrations. Indeed, **combining this method's generality and scalability makes the industrial production of multifunctional MOF fabrics possible**, leading to broad applications in the textile industry and our daily lives.

As such, we would like to **highlight the key advances** that we have made in this work:

- Such a method enabled the scalable manufacturing of functional MOFs-textiles with high durability. A roll of ZIF-67-CT textile (0.25 m wide and 30 m long) was produced.
- The covalent bonding between ZIF-67 and carboxyl chains of cellulose fibers endowed this ZIF-67-textile highly stable in air and water. After laundering based on the international standard ISO 105 C10, no obvious changes in color and crystal structure or decreased integrity of the ZIF-67-textile were observed.
- The prepared MOFs-textiles offered excellent UV resistance, antibacterial, fragrance encapsulation (for example, Carvacrol), and wastewater purification (for example, MB, RhB, and MO dye solution). For example, the UPF was up to 71 higher than 50, and the

antibacterial efficiency against *E. coli* and *S. aureus* was 99.99%, showing great potential in household products and industrial use.

- The as-prepared superhydrophobic MOFs-textiles would be widely used in outdoor products and public facilities with self-cleaning, antifouling, anti-icing, and oil-water separation properties. The total frozen time of superhydrophobic ZIF-67-CT was delayed from 90.5 s to 223.7 s, and the separation efficiency was up to 98.6% and still maintained above 97% after 15 cycles of the separation process.
- The fabrication of ZIF-67-CT is facile and scalable, and can also be applied to other cellulose-based fibers (for example, linen and paper) and MOFs (for example, ZIF-67, ZIF-8, UiO-66-NH₂, and MOF-303), hence more valuable functions would be enabled for the practical applications of textiles, including improved environmental remediation, public health, and personal protective gears.

- R4. DeCoste JB, Peterson GW. Metal–organic frameworks for air purification of toxic chemicals. *Chem Rev* **114**, 5695-5727 (2014).
- R5. Lee J, Farha OK, Roberts J, Scheidt KA, Nguyen ST, Hupp JT. Metal–organic framework materials as catalysts. *Chem Soc Rev* **38**, 1450-1459 (2009).
- R6. Howarth AJ, Liu Y, Li P, Li Z, Wang TC, Hupp JT, *et al.* Chemical, thermal and mechanical stabilities of metal–organic frameworks. *Nat Rev Mater* **1**, 1-15 (2016).
- R7. Zhu H, Yang X, Cranston ED, Zhu S. Flexible and porous nanocellulose aerogels with high loadings of metal–organic-framework particles for separations applications. *Adv Mater* **28**, 7652-7657 (2016).
- R8. Kalaj M, Bentz KC, Ayala Jr S, Palomba JM, Barcus KS, Katayama Y, *et al.* MOF-polymer hybrid materials: From simple composites to tailored architectures. *Chem Rev* **120**, 8267-8302 (2020).
- R9. Ma K, Idrees KB, Son FA, Maldonado R, Wasson MC, Zhang X, *et al.* Fiber composites of metal–organic frameworks. *Chem Mater* **32**, 7120-7140 (2020).
- R10. Peterson GW, Lee DT, Barton HF, Epps III TH, Parsons GN. Fibre-based composites from the integration of metal–organic frameworks and polymers. *Nat Rev Mater* **6**, 605-621 (2021).

Here are some **major changes** that we have made to substantially advance the manuscript:

- 1) We have re-organized the introduction to clarify the main dimensions of novelty and to make a clear description about the generic and scalable strategy by the diazonium chemistry covalent graft modification technology.
- 2) We have re-organized **Fig. 1** by adding the compared performance factors offered by other ZIF-67 coating works.
- 3) We have re-organized **Fig. 2c** to replace the low magnification SEM image with a higher magnification SEM image to clearly exhibit the morphology of ZIF-67 on the fiber surface,

and to replace the TGA curves of CT, CT-COOH, and ZIF-67-CT (**Fig. 2i**) with the color absorption spectra.

- 4) We have revised the signal marked of OH and SO⁴⁻ radicals in the EPR test results of ZIF-67-CT (**Fig. 4d**), and confirmed that the OH radical is the dominant contributor to the dye degradation, not the SO⁴⁻ radical.
- 5) We have revised **Figs. 4f, 4g, and 4h** by following the referees' comments to identify the relationship between fiber treatment methods and fiber mechanical properties.
- 6) We have added the N₂ adsorption–desorption isotherm BET of ZIF-67 powders in **Supplementary Fig. S10**, and calculated that the MOF coating content of ZIF-67-CT was 12.7% based on the normalized BET surface area calculation.
- 7) We have added the time-dependent chemical stability test data in **Supplementary Fig. S16**, and provided the full analysis process in **Supplementary Note 5**.
- 8) We have added the dye degradation kinetic data in **Supplementary Figs. S25 and S26**, and provided the full analysis process in **Supplementary Note 7**.
- 9) We have added the SEM images of the washed ZIF-67-CT at varied locations to prove the laundering stability of ZIF-67-CT in **Supplementary Fig. S27**.
- 10) We have added the N₂ adsorption–desorption isotherm BET of the washed sample in **Supplementary Fig. S29**, and the residual MOF content after the washing testing is calculated to be ~ 10.6% based on the changes in BET surface area (the MOF content before the washing testing is ~ 12.7%).
- 11) We have added the fabrication of UiO-66-NH₂-CT and MOF-303-CT via in situ growth of UiO-66-NH₂ and MOF-303 on carboxymethylated cotton textiles in **Supplementary Figs. S36 and S37**, and provided the analysis process in **Supplementary Note 8**.

1. Please add the page number to the manuscript.

Response:

We thank the referee for the comment. We have added the page number in the revised manuscript.

2. (Line 50-54 on page 2) Relevant references are absent concerning dip-coating, hot-pressing, and spraying methods for MOF integration, although those are stated in the text.

Response:

We are really grateful for this comment. We have added the relevant references as refs [33-35] in the revised manuscript. “current mainstream strategies for fabricating MOFs-fibers are typically incorporated into the fibers with non-covalent bond using methods such as dip-coating, hot-pressing, and spraying [33-35].”

33. Zhang Y, Yuan S, Feng X, Li H, Zhou J, Wang B. Preparation of nanofibrous metal–organic framework filters for efficient air pollution control. *J Am Chem Soc* **138**, 5785-5788 (2016).
34. Chen Y, Li S, Pei X, Zhou J, Feng X, Zhang S, *et al.* A solvent - free hot - pressing

method for preparing metal – organic – framework coatings. *Angew Chem Int Ed* **55**, 3419-3423 (2016).

35. Li C, Zhang Q, Sun J, Li T, E S, Zhu Z, *et al.* High-performance quasi-solid-state flexible aqueous rechargeable Ag–Zn battery based on metal–organic framework-derived Ag nanowires. *ACS Energy Letters* **3**, 2761-2768 (2018).

3. (Line 54-57 on page 2) It seems that the authors claim that the references (#30-#34) only demonstrate the physical incorporation of MOFs into polymeric fiber substrates, but it is not... The key to some of the work the authors of this work refer to is associated with the chemical integration of MOFs into textiles. Please double-check the references and replace them (not related) with relevant ones.

Response:

Thanks for this comment, and we have carefully double-checked the references (Ref [30] - Ref [34]).

Ref [30] shows that nonwoven fiber mats were coated with Al₂O₃ by atomic layer deposition (ALD) using a homemade hot-wall viscous-flow vacuum reactor, followed by layer-by-layer MOF growth (**Atomic layer deposition coating**).

Ref [31] shows that MOF-808 growth on fiber is via a three-stage process: (1) aggregation of an amorphous coordination polymer on the fibrous surface (0–10 min), (2) formation of MOF-808 nucleation sites (10–20 min), and (3) growth into a continuous MOF coating (>30 min). It is inferred that this period served as a pseudotemplating period during which reagents were adsorbed on the surface of the fiber and quickly formed an amorphous coordination polymer (**Adsorption aggregation on the fiber surface**).

Ref [32] shows that TiO₂ coatings were deposited via atomic layer deposition (ALD) onto polyamide-6 nanofibers, enabling the formation of conformal Zr-based MOF thin films, including UiO-66, UiO-66-NH₂, and UiO-67 (**Atomic layer deposition coating**).

Ref [33] shows that the synthetic procedure of MOF-808/PEI/fiber composite by being immersed into the suspension for 10 min, and then taken out from suspension. The wet fabric with 200% liquid pick-up was carefully put on an aluminium foil and dried in a hood overnight. (**Impregnation method**).

Ref [34] shows that the nanofibrous metal–organic framework filters were prepared by electrospinning with the mixture of MOFs and PAN (**Mixing method**).

These mentioned references do not have a chemical covalent bond force between fibers and the MOF coating. We deleted ref [30], as it is similar to ref [32]. The relevant references have been reorganized to be refs [29-32], due to the re-written introduction of the revised manuscript.

29. Ma K, Islamoglu T, Chen Z, Li P, Wasson MC, Chen Y, *et al.* Scalable and template-free aqueous synthesis of zirconium-based metal–organic framework coating on textile fiber. *J Am Chem Soc* **141**, 15626-15633 (2019).

30. Zhao J, Lee DT, Yaga RW, Hall MG, Barton HF, Woodward IR, *et al.* Ultra-fast degradation of chemical warfare agents using MOF–nanofiber kebabs. *Angew Chem* **128**, 13418-13422 (2016).
31. Chen Z, Ma K, Mahle JJ, Wang H, Syed ZH, Atilgan A, *et al.* Integration of metal–organic frameworks on protective layers for destruction of nerve agents under relevant conditions. *J Am Chem Soc* **141**, 20016-20021 (2019).
32. Zhang L, Chen H, Bai X, Wang S, Li L, Shao L, *et al.* Fabrication of 2D metal–organic framework nanosheet@ fiber composites by spray technique. *Chem Commun* **55**, 8293-8296 (2019).

4. (on page 3) It is stated that “In addition, the fabrication of ZIF-67-CT is facile and scalable...” According to the preparation methods of MOF-textiles in this work, it takes more than a day, including a carboxymethylation of cotton fabric and MOF growth onto it. A more detailed explanation regarding the extent of being “facile and scalable” compared to other reported MOFs-textiles is necessary to convince the reviewers and readers.

Response:

We thank the referee for the valuable comment. The preparation of each type of MOF takes a certain amount of time to confirm its special crystal structures, for example, ZIFs MOF need more than 12 hours, supported by the following references [R11-R14].

Processing condition and time	Reference
Room temperature for 15 h	R11
358 K for 72 h	R12
Solvothermal for 24–96 h	R13
Room temperature for 12 h	R14

In our work, the carboxymethylation of cotton fabric is modified via diazonium radical graft polymerization, which can be completed from 1 to 24 h, and the grafting process time of more than 2 h is sufficient for the subsequent MOF growth. For example, the scalable textile sample shown in Fig. 5c was prepared within 3 h. More importantly, **the whole manufacturing process can be finished at room temperature, which does not need high temperature, complex fabrication process, or special equipment, just one container** (Fig. 5b). For example, a roll of ZIF-67-CT cloth (0.25 m wide and 30 m long) was easily produced using a container (diameter of 27 cm and height of 15 cm) in laboratory (Fig. 5c). Therefore, we stated “the fabrication of ZIF-67-CT is facile and scalable...”.

- R11. Jin M, Lu S-Y, Ma L, Gan M-Y, Lei Y, Zhang X-L, *et al.* Different distribution of in-situ thin carbon layer in hollow cobalt sulfide nanocages and their application for supercapacitors. *J Power Sources* **341**, 294-301 (2017).
- R12. Quiros J, Boltes K, Aguado S, de Villoria RG, Vilatela JJ, Rosal R. Antimicrobial metal–organic frameworks incorporated into electrospun fibers. *Chem Eng J* **262**, 189-197 (2015).

- R13. Banerjee R, Phan A, Wang B, Knobler C, Furukawa H, O’Keeffe M, *et al.* High-throughput synthesis of zeolitic imidazolate frameworks and application to CO₂ capture. *Science* **319**, 939-943 (2008).
- R14. Park J, Oh M. Construction of flexible metal–organic framework (MOF) papers through MOF growth on filter paper and their selective dye capture. *Nanoscale* **9**, 12850-12854 (2017).

Fig. 5 Scalable manufacturing of the ZIF-67-CT. a, The fabrication process of the ZIF-67-CT via in-situ growth. b, Photographs showing the fabrication process of ZIF-67-CT. c, Photographs of the large-scale ZIF-67-CT (30 m × 0.25 m).

5. (Line 99-101 on page 4) It is stated that “Scanning electron microscopy (SEM) micrographs show that lots of ZIF-67 particles are evenly distributed on the carboxymethylated fiber surface, forming a MOF coating (Fig. 2c).” The resolution of the SEM image (Fig. 2c, right) is not good enough to tell the morphology of ZIF-67 on the fiber surface. Taking a higher magnification image of the sample is recommended. In addition, Fig. S6 and S7 do not agree with the text. Please revise it.

Response:

We thank the referee for the valuable comment. As suggested by the referee, we have replaced Fig. 2c (right) with a higher magnification SEM image to exhibit the morphology of ZIF-67 on the fiber surface. In addition, we are very sorry for our mistake about Figs. S6 and S7, and we have corrected the corresponding text in the revised manuscript.

Fig. 2c (revised) SEM images of ZIF-67-CT with different magnifications.

6. (Line 116-117 on page 4) *The authors claim that “as observed in Fig. 2e, the results from XPS indicate the presence of carbon, oxygen, nitrogen and cobalt species in ZIF-67-CT, which are well in agreement with EDS results.” In this work, the authors have only considered the presence of the major elements comprising ZIF-67. However, it is recommended that the authors maximize the utilization of the obtained XPS data by calculating element area ratios from XPS for comparison with those of ZIF-67 synthesized in the liquid phase.*

Response:

We thank the referee for the valuable comment. As suggested by the referee, we have calculated the element atomic (%) and area ratios of ZIF-67-CT and ZIF-67 from the XPS test reports (Table R1). The atomic (%) and area ratio C:O:N:Co of prepared ZIF-67-CT is 14.9:4.1:2.4:1 and 3.9:2.6:1:3.2, respectively. In contrast, as there is no oxygen element in ZIF-67, the atomic (%) and area ratio (C:N:Co) of ZIF-67 synthesized in liquid phase is 8.9:3.7:1 and 3.8:1:3.1, respectively. Besides, we have calculated the ZIF-67 content, which is 12.7%, according to the normalized BET surface area calculation.

Table R1. Comparison between the element atomic (%) and area ratios of ZIF-67-CT and ZIF-67.

Sample	Atomic %	Area ratio
ZIF-67-CT (C:O:N:Co)	14.9:4.1:2.4:1	3.9:2.6:1:3.2
ZIF-67 (C:N:Co)	8.9:3.7:1	3.8:1:3.1

7. (Line 121-125 on page 5) *The authors claim that “The fabricated ZIF-67-CT 121 and ZIF-67 have the same crystal characteristic peaks, consistent well with the simulated crystal pattern of ZIF-67...” As shown in Fig. 2g (on page 6), It appears that there are shifts in the major peaks corresponding to ZIF-67 among ZIF-67-CT, ZIF-67 powder, and simulated ZIF-67. In addition, it is highly recommended that the authors calculate the specific values of the lattice parameters, peak broadness, and peak intensity ratios of the ZIF-67 component on the fabric to investigate any structural differences compared to freestanding ZIF-67 crystal powder.*

Response:

We thank the referee for the valuable comment. As shown in Fig. 2g, the characteristic peaks of ZIF-67 appear in the XRD patterns of ZIF-67-CT and ZIF-67 powder, which are the same as ZIF-67 simulated, indicating the successful synthesis of ZIF-67 crystal in this work. The slight shift in the

major peaks is mainly caused by the difference in the sample state during the test processing. As known, the sample status, sample preferential orientation, and testing instrument model would affect the peak shift (for example, *Angew. Chem. Int. Ed.* 2016, 55, 3419–3423).

As for the comment “*In addition, it is highly recommended that the authors calculate the specific values of the lattice parameters, peak broadness, and peak intensity ratios of the ZIF-67 component on the fabric to investigate any structural differences compared to freestanding ZIF-67 crystal powder.*” As shown in Fig. R1, the ZIF-67-CT has the crystal characteristic peaks of ZIF-67 at the same 2 theta positions with freestanding ZIF-67 powder, which demonstrated that the ZIF-67 component on the fabric and freestanding ZIF-67 powder has same lattice parameters. However, the main difference between the ZIF-67 component on the fabric and freestanding ZIF-67 powder is peak broadness and peak intensity, which represent the content of the ZIF-67 crystal structure in the tested samples. The freestanding ZIF-67 powder has 100% of ZIF-67 crystal, whereas the ZIF-67 crystal content of ZIF-67-CT is measured to be 12.7%, according to the normalized BET surface area calculation. Therefore, the intensity of freestanding ZIF-67 crystal powder is higher than ZIF-67-CT. As shown in Table R2, the peak width at half height of ZIF-67-CT and ZIF-67 crystal powder is 0.257 and 0.265, respectively. And the peak intensity ratio of ZIF-67 component on the fabric and freestanding ZIF-67 crystal powder is 0.2865 at the main peak (2 Theta=7.2) in Table R2.

In addition, the BET is the main method to identify the micropores and specific surface area of MOFs. Moreover, we combined the ATR-FTIR, XPS, and SEM tests to confirm the structures of MOFs. Thus, we believe these results are sufficient to support our claims regarding the prepared ZIF-67-CT with great ZIF-67 crystal structures in this work. Besides, other published papers about ZIF-67 or other MOFs@fiber composites can support our conclusion, as refs [R15-R21].

- R15. Quiros J, Boltes K, Aguado S, de Villoria RG, Vilatela JJ, Rosal R. Antimicrobial metal–organic frameworks incorporated into electrospun fibers. *Chem Eng J* **262**, 189-197 (2015).
- R16. Park J, Oh M. Construction of flexible metal–organic framework (MOF) papers through MOF growth on filter paper and their selective dye capture. *Nanoscale* **9**, 12850-12854 (2017).
- R17. Zhang S, Zhao M, Li H, Hou C, Du M. Facile in situ synthesis of ZIF-67/cellulose hybrid membrane for activating peroxydisulfate to degrade organic contaminants. *Cellulose* **28**, 3585-3598 (2021).
- R18. Qiao X, Gao W, Liu X, Fang K, Li Q, Lu X, *et al.* Preparation of zeolitic imidazolate framework-67/wool fabric and its adsorption capacity for reactive dyes. *Journal of Environmental Management* **321**, 115972 (2022).
- R19. Chen Y, Li S, Pei X, Zhou J, Feng X, Zhang S, *et al.* A solvent-free hot-pressing method for preparing metal–organic-framework coatings. *Angew Chem Int Ed* **55**, 3419-3423 (2016).

- R20. Ma K, Cheung YH, Kirlikovali KO, Xie H, Idrees KB, Wang X, *et al.* Fibrous Zr-MOF Nanozyme Aerogels with Macro-Nanoporous Structure for Enhanced Catalytic Hydrolysis of Organophosphate Toxins. *Adv Mater* 2300951 (2023).
- R21. Zhang L, Chen H, Bai X, Wang S, Li L, Shao L, *et al.* Fabrication of 2D metal–organic framework nanosheet@ fiber composites by spray technique. *Chem Commun* **55**, 8293-8296 (2019).

Fig. 2g The XRD patterns of CT, ZIF-67-CT, ZIF-67 powder, and simulated ZIF-67.

Fig. R1 The XRD patterns of ZIF-67-CT and freestanding ZIF-67 powder after background subtraction.

Table R2. Peak width at half height and peak intensity of ZIF-67 component on the fabric and freestanding ZIF-67 crystal powder.

Sample	Peak width at half height	Peak intensity
ZIF-67 component on the fabric	0.257	1387
Freestanding ZIF-67 crystal powder	0.265	4841

8. (Line 157-159 on page 7) The authors state that “the modified ZIF-67-CT exhibits excellent superhydrophobic and superoleophilic properties and shows good acid and alkali resistance stability (Supplementary Fig. S15).” Time-dependent chemical stability data for modified ZIF-67-CT should be conducted and included in this work to claim the point above.

Response:

We thank the referee for the valuable comment. As suggested by the referee, we have added the time-dependent chemical stability test data. As shown in Supplementary Fig. S16, the ZIF-67-CT was sunk to the bottom of the acid and alkali solutions (Figs. S16a and S16c, left), and the color of ZIF-67-CT was transformed from purple into a faint yellow after soaking for 48 h at 30 °C. On the contrary, the modified superhydrophobic ZIF-67-CT completely floated on the water in the bottle (Figs. S16a and S16c, right) due to the superhydrophobicity, and the color of the superhydrophobic ZIF-67-CT was not changed even after soaking under the same conditions. The ZIF-67 crystal structure of ZIF-67-CT was lost after soaking for 48 h (Figs. S16b and S16d), whereas the characteristic peaks of the modified superhydrophobic ZIF-67-CT were well maintained. The time-dependent chemical stability test data results demonstrate that the superhydrophobic modification of ZIF-67-CT shows good acid and alkali resistance stability.

Supplementary Fig. S16 Photographs of ZIF-67-CT (left) and modified superhydrophobic ZIF-67-CT (right) after exposure to a pH=2 (a) and pH=13 (c) solution for 48 h at 30 °C. The corresponding

XRD patterns of ZIF-67-CT and modified superhydrophobic ZIF-67-CT after exposure to a pH=2 (b) and pH=13 (d) solution.

9. (Line 170-172 on page 7) The authors mention that “...which may be due to the transition of electrons[36] of the Co^{2+} of ZIF-67.” More detailed but concise information regarding the speculation is required to be included.

Response:

We thank the referee for the valuable comment. We have added concise information about the ultraviolet mechanism in the revised manuscript, and the anti-ultraviolet mechanism of the ZIF-67-CT is explained as follows. Because metallic oxide is a good inorganic UV blocker, the UV absorption of MOF coating stems from the transition of electrons, which can be supported by the following references [R22 and R23]. Specifically, when there is no external stimulus light, the electrons of metal in the MOF stay in the valence band of low energy. Once the ZIF-67 is irradiated by the incident light, it absorbs the UV photonic energy, enabling the transition of electrons from the valence band to the conduction band with high energy. Thus, the ZIF-67-CT exhibits excellent UV protection performance.

R22. Shi YE, Zhuang X, Cao L, Gou S, Xiong Y, Lai WF, *et al.* Copper-Nanocluster-Based Transparent Ultraviolet-Shielding Polymer Films. *ChemNanoMat* **5**, 110-115 (2019).

R23. Li W, Liu K, Zhang Y, Guo S, Li Z, Tan SC. A facile strategy to prepare robust self-healable superhydrophobic fabrics with self-cleaning, anti-icing, UV resistance, and antibacterial properties. *Chem Eng J* **446**, 137195 (2022).

10. (Line 221-223 on page 9) The authors claim that “... the ZIF-67-CT exhibits superior antibacterial activity against bacteria...” I understand that the ZIF-67-CT shows antibacterial activity through the data provided in the manuscript. Still, as a reader, I cannot tell how effective the ZIF-67-CT is in comparison with other MOF-textile samples and related materials. Please make a good comparison with other materials in the antibacterial activity.

Response:

We are really grateful for this comment. As we stated in the manuscript “We calculated the antibacterial efficiency using the plate count method, and found the antibacterial efficiency of ZIF-67-CT and ZIF-67-CT/Carvacrol against *E. coli* and *S. aureus* was 99.99% (Supplementary Fig. S21).” When the antibacterial efficiency of one sample is above 90%, which can be defined as the sample with great antibacterial activity. In this work, the antibacterial efficiency of the prepared sample is up to 99.99%, showing superior antibacterial activity against *E. coli* and *S. aureus*. And this can be further compared with other refs [R24-R26]. In addition, the inhibition zones of most MOFs are typically within 0.5-20.0 mm, and the summary of the antibacterial action of selected MOFs can be found in ref [R27]. While, the inhibition zone of our prepared ZIF-67-CT/Carvacrol inoculated *E. coli* is obviously increased to 33.0 mm, especially for *S. aureus* (90.0 mm), significantly larger than 20.0 mm (Supplementary Fig. S20). Therefore, combining the disc-diffusion analytic method and plate count method results, the ZIF-67-CT exhibits superior antibacterial activity, especially the carvacrol-loaded ZIF-67-CT.

- R24. Wang F, Yan B, Li Z, Wang P, Zhou M, Yu Y, *et al.* Rapid antibacterial effects of silk fabric constructed through enzymatic grafting of modified PEI and AgNP deposition. *ACS Appl Mater Interfaces* **13**, 33505-33515 (2021).
- R25. Lis MJ, Caruzi BB, Gil GA, Samulewski RB, Bail A, Scacchetti FAP, *et al.* In-situ direct synthesis of HKUST-1 in wool fabric for the improvement of antibacterial properties. *Polymers* **11**, 713 (2019).
- R26. Li P, Li J, Feng X, Li J, Hao Y, Zhang J, *et al.* Metal-organic frameworks with photocatalytic bactericidal activity for integrated air cleaning. *Nat Commun* **10**, 2177 (2019).
- R27. Wyszogrodzka G, Marszałek B, Gil B, Dorożyński P. Metal-organic frameworks: mechanisms of antibacterial action and potential applications. *Drug Discovery Today* **21**, 1009-1018 (2016).

Supplementary Fig. S20 Inhibition zones of CT, ZIF-67-CT, and ZIF-67-CT/Carvacrol against *E. coli* and *S. aureus*.

Supplementary Fig. S21 The antibacterial efficiency of CT, ZIF-67-CT, and ZIF-67-CT/Carvacrol against *E. coli* and *S. aureus*.

11. (Line 230-232 on page 9) The authors claim that "... the ZIF-67-CT exhibited excellent catalytic performance." I cannot find any dye degradation kinetic data (dye concentration vs. time) in the main text as well as in the supplementary document. Please provide the kinetic data.

Response:

We thank the referee for the valuable comment. As suggested by the referee, we have added the dye degradation kinetic data in the revised manuscript, as shown in Supplementary Fig. S25, Fig. S26, and Supplementary Note 7: The degradation ability. We first investigated the influence of the PMS concentration on the ZIF-67-CT catalytic degradation. Supplementary Fig. S25 shows the degradation performance of the methylene blue solution with six different PMS concentrations (0.0, 0.6, 0.9, 1.2, 1.5, and 1.8 mg/mL) before and after degradation by ZIF-67-CT at room temperature, indicating that higher concentration of the PMS corresponds to more colorless of the solution, and then showing more degradation effect. As a result, 98.2% degradation efficiency was reached at 1.8 mg/mL of PMS. In addition, we investigated the dye degradation kinetic of ZIF-67-CT with 1.8 mg/mL of PMS at room temperature. Supplementary Fig. S26 shows that the ZIF-67-CT could degrade 96.8% of the MB within 8 min, and slightly increased to 99.1% in 10 min. Therefore, we used the 1.8 mg/mL PMS to examine the degradation property of ZIF-67-CT on rhodamine B (RhB) and methyl orange (MO) solutions.

Supplementary Fig. S25 The degradation performance of the methylene blue solution with different concentrations of PMS before and after degradation by ZIF-67-CT at room temperature (MB solution = 50 mg/L, PMS = 1.8 mg/L, pH = 7).

Supplementary Fig. S26 The MB removal rate at room temperature (MB solution = 50 mg L⁻¹, PMS = 1.8 mg/mL, pH = 7).

12. (Line 245-247 on page 10) The authors claim that “The SEM images of the ZIF-67-CT after washing show the same morphology structure as before washing...” However, there appears to be a noticeable change before and after the fabric composite washing (Fig. 4e). It is recommended that the authors take SEM images of the washed samples at varied locations. I do not think the SEM images in Fig. S24 represent the samples after washing.

Response:

We thank the referee for the valuable comment. As suggested by the referee, we have re-taken SEM images of the washed ZIF-67-CT at various locations and added the SEM images in the revised Supplementary Fig. S27. SEM images of the ZIF-67-CT after laundering show that some MOF particles are off the fiber surface at some locations (such as Location 1), but most MOF coatings are well maintained (such as Location 2), demonstrating that ZIF-67-CT shows acceptable laundering stability. In addition, we have also tested the corresponding XRD spectrum at these two locations after laundering (Supplementary Fig. S28), and ZIF-67 crystal characteristic peaks of ZIF-67-CT after laundering matched well with before laundering. Furthermore, when the 2 theta is less than 15, the ZIF-67 crystal characteristic peaks intensity of location 1 weaker than location 2, whereas the cellulose crystal characteristic peaks intensity of location 1 stronger than location 2, which is due to more MOF particles maintained on the fiber surface of location 2 than location 1. This result is consistent with SEM results. Above all, the intensity of ZIF-67 characteristic peaks of location 1 is slightly weaker than location 2, and location 2 is weaker than before laundering due to the MOF content being decreased after laundering and consistent with the SEM results.

Supplementary Fig. S27 (revised) The photograph and SEM images of ZIF-67-CT after laundering at various locations.

Supplementary Fig. S28 (revised) The XRD spectrum of ZIF-67-CT before and after laundering at various locations.

13. (Line 250-252 on page 10) The authors state that “... the fiber structure damage of ZIF-67-CT during the modification processes...” Here, the authors should characterize CT-COOH and ZIF-67-CT after washing to confirm what causes the fiber structure damage of ZIF-67-CT, and to identify the relationship between fiber treatment methods and fiber mechanical properties.

Response:

We thank the referee for the valuable comment. As known, the mechanical strength of cotton textile decreases when it is treated with acid, alkaline, or organic solutions, which may be attributed to the decrease in the degree of polymerization of the cellulose polymers. As shown in Fig. 4f (revised), the tensile strength of CT-COOH is 26.28 MPa, similar to ZIF-67-CT (25.98 MPa), which is lower than that of the pristine cotton textile CT (31.43 MPa), demonstrating that the decrease of the tensile strength of cotton fiber is mainly caused by the acid solution treatment during the carboxymethylation of CT in HCl solution. The following references can support this conclusion [R28-R32]. As shown in Fig. 4g and 4h (revised), the flexural rigidity and air permeability of CT-

COOH are similar to CT, indicating that the carboxymethylation process would not influence its flexural rigidity and air permeability performances. In addition, compared to CT-COOH, the air permeability of ZIF-67-CT is decreased from 258.6 mm s⁻¹ to 203.5 mm s⁻¹ (Fig. 4h), attributed to the loaded MOF coating on the fiber surface of ZIF-67-CT.

- R28. Qian J, Dong Q, Chun K, Zhu D, Zhang X, Mao Y, *et al.* Highly stable, antiviral, antibacterial cotton textiles via molecular engineering. *Nat Nanotechnol* **18**, 168-176 (2023).
- R29. Knill CJ, Kennedy JF. Degradation of cellulose under alkaline conditions. *Carbohydr Polym* **51**, 281-300 (2003).
- R30. de Carvalho Benini KC, Pereira PH, Cioffi MOH, Voorwald HJC. Effect of acid hydrolysis conditions on the degradation properties of cellulose from *Imperata brasiliensis* fibers. *Procedia Engineering* **200**, 244-251 (2017).
- R31. Garves K. Acid catalyzed degradation of cellulose in alcohols. *J Wood Chem Technol* **8**, 121-134 (1988).
- R32. Wiyantoko B, Rusitasari R, Putri RN. Study of Hydrolysis Process from Pineapple Leaf Fibers using Sulfuric Acid, Nitric Acid, and Bentonite Catalysts. *Bulletin of Chemical Reaction Engineering & Catalysis* **16**, 571-580 (2021).

Fig. 4f-h (revised) (f) Tensile strength curves, (g) flexural rigidity, and (h) air permeability of CT, CT-COOH, and ZIF-67-CT.

14. (Line 255-257 on page 10) The authors claim that “These results demonstrate no obvious change in physical-mechanical properties, which does not affect its daily use.” In order to claim that point, the authors should show N₂ isotherms of ZIF-67-CT after washing and compare with those of as-prepared ZIF-67-CT. Surface area, pore volume, and porosity are closely related to MOF-textiles properties, so maintaining such properties after laundering stability tests is critical in this work.

Response:

We thank the referee for the valuable comment. We have added the N₂ adsorption-desorption isotherm BET and the pore width distribution of the washed sample. As shown in Supplementary Fig. S29, the BET surface area of the washed sample is 207.6 m² g⁻¹, and the pore width of the mesoporous structure is mainly at ≈ 0.64 nm. Compared to ZIF-67-CT before laundering (244.2 m² g⁻¹), the BET surface area is decreased by about 15%, mainly because some MOF particles are off the fiber surfaces during the laundering process. This result indicated that the laundering process

did not largely impact the stability of the MOF coating. In addition, the residual MOF content of ZIF-67-CT after washing testing is calculated to be ~10.6% based on the changes in BET surface area (the MOF content before the washing testing is ~12.7%).

Supplementary Fig. S29 (a) N₂ adsorption–desorption isotherm and (b) pore width distribution of ZIF-67-CT after laundering.

15. (Line 307-309 on page 13) The authors conclude that “Besides ZIF-67 MOF and cotton textiles, our developed strategy can also be applied to other MOFs (for example, ZIF-8 MOF)...” In my humble opinion, demonstrating ZIF-8 integration is not enough to claim the method is generic. The authors need to show other MOFs beyond the ZIFs with SOD (sodalite-like) structures.

Response:

We thank the referee for the valuable comment. As suggested by the referee, in order to better prove that the method is generic, we have prepared other MOF textiles, such as zirconium-based MOF textile UiO-66-NH₂-CT (Supplementary Fig. S36) and aluminum-based MOF textile MOF-303-CT (Supplementary Fig. S37). And the crystal structures of the fabricated UiO-66-NH₂-CT and MOF-303-CT were confirmed using SEM and XRD (Supplementary Note 8).

Supplementary Fig. S36 The (a) photograph and (b) SEM images of UiO-66-NH₂-CT; (c) 3D

simulated chemical structure of UiO-66-NH₂; (d) XRD patterns of CT, UiO-66-NH₂-CT and simulated UiO-66-NH₂.

Supplementary Fig. S37 The (a) photograph and (b) SEM images of MOF-303-CT; (c) 3D simulated chemical structure of MOF-303; (d) XRD patterns of CT, MOF-303-CT and simulated MOF-303.

REVIEWER COMMENTS

Reviewer #1 (Remarks to the Author):

The authors carefully addressed my questions and fixed the majority of the issues. I recommend publishing the paper after minor revision.

Could you explain why the N₂ sorption capacity of ZIF-67-CT is improved after the laundering process? Please double-check Supplementary Fig. S29a and Figure 2h.

Reviewer #4 (Remarks to the Author):

This paper describes the preparation of ZIF-67 composite with cotton textile using diazonium chemistry to anchor the MOF onto the surface of the textile. The authors use MOF-coated textiles to encapsulate essential oils and demonstrate the utility of this composite material as an antibacterial agent. The methodology appears to be highly scalable and durable, leading to meter-scale functionalization. Overall, the materials are well characterized, and the methodology has unique advantages over existing methodologies. I recommend several minor revisions before publication:

1. Have the authors confirmed MOF loading on the textile using a technique other than BET? While BET may provide a reasonable estimate, a more quantitative approach based on ICP-MS or another technique could be valuable. Also, can the authors control MOF loading through the density of covalent surface modification?
2. The authors should acknowledge other important contributions of integrating MOFs onto textiles to place their method into the context of the field. For instance, a recent review on the topic by Eagleton et al. is recommended to be cited: <https://onlinelibrary.wiley.com/doi/10.1002/anie.202309078>. The authors should also acknowledge self-assembly based methods for integrating MOFs into textiles, such as one reported by Smith et al. (<https://pubs.acs.org/doi/full/10.1021/jacs.7b08840>). Furthermore, some robust strategies for MOF integration into textiles have been developed, and should be acknowledge: <https://pubs.acs.org/doi/full/10.1021/jacs.2c05510>

Point-by-Point Responses to Referees' Comments

We thank all the referees for their in-depth review of our manuscript and for enriching us with their valuable comments, which helped us further improve the quality of the manuscript. The referees' comments are listed in *blue font color*, and the authors' responses are listed in black font color. All the changes in the revised manuscript and supplementary information are marked in *red font color*.

Referee #1:

The authors carefully addressed my questions and fixed the majority of the issues. I recommend publishing the paper after minor revision.

Response:

We really appreciate your in-depth review of our manuscript and encouraging feedback. According to your comments and suggestions, we have further revised and improved our manuscript.

Could you explain why the N₂ sorption capacity of ZIF-67-CT is improved after the laundering process? Please double-check Supplementary Fig. S29a and Figure 2h.

Response:

To confirm the result of the N₂ sorption capacity of ZIF-67-CT before and after the laundering process, we have double-tested the N₂ adsorption–desorption isotherm of ZIF-67-CT after the laundering process. We have updated the N₂ adsorption–desorption isotherm BET of the washed sample in Supplementary Fig. S29. The BET surface area is 202.9 m² g⁻¹ (after the laundering), which is less than 244.2 m² g⁻¹ (before the laundering), and the pore width of the mesoporous structure is mainly at ≈ 0.62 nm. The N₂ sorption capacity of ZIF-67-CT is decreased after the laundering process, which is mainly due to the loss of MOF during the laundering process.

In addition, we tested the ZIF-67 loading before and after the laundering process on fibers, and the results were calculated to be 11.4% and 10.1% based on ICP-OES analysis, respectively. The MOF mass loading (MML) is calculated using the following equation:

$$\text{MML (\%)} = (C_1 \times V_0) / (m_0 \times W_{\text{Co}} \times 10^6) \times 100\%$$

where C_1 is the concentration of Co in the diluted nitric acid solution measured by ICP-OES in mg/L, V_0 is the volume of the test solution in mL, m_0 (g) is the quality of the test sample, and W_{Co} is the mass percentage in MOFs.

Supplementary Fig. S29 (a) N₂ adsorption–desorption isotherm and (b) pore width distribution of ZIF-67-CT after the laundering process.

Referee #4:

This paper describes the preparation of ZIF-67 composite with cotton textile using diazonium chemistry to anchor the MOF onto the surface of the textile. The authors use MOF-coated textiles to encapsulate essential oils and demonstrate the utility of this composite material as an antibacterial agent. The methodology appears to be highly scalable and durable, leading to meter-scale functionalization. Overall, the materials are well characterized, and the methodology has unique advantages over existing methodologies. I recommend several minor revisions before publication:

Response:

We really appreciate your in-depth review of our manuscript. According to your comments and suggestions, we have revised and improved our manuscript.

1. Have the authors confirmed MOF loading on the textile using a technique other than BET? While BET may provide a reasonable estimate, a more quantitative approach based on ICP-MS or another technique could be valuable. Also, can the authors control MOF loading through the density of covalent surface modification?

Response:

We thank the referee for the valuable comment. In the original manuscript, we tested the MOF loading on the textile using TGA analysis, which had a larger variation compared with its real MOF loading. This is because the residual after calcining of MOF is oxides, not ZIF-67. As suggested by the referee, we have added the normalized BET surface area calculation and ICP-OES/MS methods to test the MOF coating content of ZIF-67-CT. And the ZIF-67 loading on fiber is calculated to be 11.4% based on ICP-OES analysis, which is similar to the result based on BET surface area calculation (12.7%). The MOF mass loading (MML) is calculated using the following equation:

$$\text{MML (\%)} = (C_1 \times V_0) / (m_0 \times W_{Co} \times 10^6) \times 100\%$$

where C_1 is the concentration of Co in the diluted nitric acid solution measured by ICP-OES in mg/L, V_0 is the volume of the test solution in mL, m_0 (g) is the quality of the test sample, and W_{Co} is the mass percentage in MOFs.

In addition, the MOF loading could be increased with the increase of covalent surface modification density, which can be controlled by the covalent surface modification time. Hence the MOF loading could be controlled within a certain covalent surface modification time range.

The additional information has been included in the revised manuscript.

2. The authors should acknowledge other important contributions of integrating MOFs onto textiles to place their method into the context of the field. For instance, a recent review on the topic by Eagleton et al. is recommended to be cited: <https://onlinelibrary.wiley.com/doi/10.1002/anie.202309078>. The authors should also acknowledge self-assembly based methods for integrating MOFs into textiles, such as one reported by Smith et al. (<https://pubs.acs.org/doi/full/10.1021/jacs.7b08840>). Furthermore, some robust strategies for MOF integration into textiles have been developed, and should be acknowledge: <https://pubs.acs.org/doi/full/10.1021/jacs.2c05510>

Response:

We thank the referee for providing these valuable references for us to understand the important contributions of integrating MOFs onto textiles. These references have been included in the revised

manuscript as refs [11], [15] and [31].

11. Eagleton A. M. *et al.* Fiber integrated metal - organic frameworks as functional components in smart textiles. *Angew. Chem. Int. Ed.* **135**, e202309078 (2023).
15. Smith M. K. & Mirica K. A. Self-organized frameworks on textiles (SOFT): conductive fabrics for simultaneous sensing, capture, and filtration of gases. *J. Am. Chem. Soc.* **139**, 16759-16767 (2017).
31. Eagleton A. M. *et al.* Fabrication of multifunctional electronic textiles using oxidative restructuring of copper into a Cu-based metal–organic framework. *J. Am. Chem. Soc.* **144**, 23297-23312 (2022).

REVIEWERS' COMMENTS

Reviewer #1 (Remarks to the Author):

The authors carefully fixed the issues. The newly added ICP method helps evaluate the MOF mass loading before and after the washing test. I recommend this paper for publication.

Response to reviewer's comments:

The reviewer's comment is in *italic*, and the authors' response is in **Roman**.

Reviewer #1

The authors carefully fixed the issues. The newly added ICP method helps evaluate the MOF mass loading before and after the washing test. I recommend this paper for publication.

Response:

We are glad that our response has addressed the reviewer's comment. And we would like to thank the reviewer for the helpful comments.